# Electrospun Fibers Loaded with Natural Bioactive Compounds as a Biomedical System for Skin Burn Treatment. A Review

**DOI:** 10.3390/pharmaceutics13122054

**Published:** 2021-12-01

**Authors:** Jeyson Hermosilla, Edgar Pastene-Navarrete, Francisca Acevedo

**Affiliations:** 1Doctoral Program in Sciences of Natural Resources, Universidad de La Frontera, Casilla 54-D, Temuco 4780000, Chile; j.hermosilla03@ufromail.cl; 2Laboratorio de Síntesis y Biotransformación de Productos Naturales, Universidad del Bío-Bío, Chillán 3800708, Chile; epastene@ubiobio.cl; 3Department of Basic Sciences, Faculty of Medicine, Universidad de La Frontera, Casilla 54-D, Temuco 4780000, Chile; 4Center of Excellence in Traslational Medicine (CEMT), Faculty of Medicine, and Scientific and Technological Bioresource Nucleus (BIOREN), Universidad de La Frontera, Casilla 54-D, Temuco 4780000, Chile

**Keywords:** antimicrobial agents, electrospinning, wound healing, biomaterials

## Abstract

Burns are a major threat to public health and the economy due to their costly and laborious treatment and high susceptibility to infection. Efforts have been made recently to investigate natural bioactive compounds with potential use in wound healing. The importance lies in the capacities that these compounds could possess both in infection control by common and resistant microorganisms, as well as in the regeneration of the affected tissues, having in both cases low adverse effects. However, some bioactive molecules are chemically unstable, poorly soluble, and susceptible to oxidative degradation or have low bioavailability. Therefore, developing new technologies for an efficient treatment of wound healing poses a real challenge. In this context, electrospun nanofibers have gained increasing research interest because bioactive molecules can be easily loaded within the nanofiber, resulting in optimal burst control and enhanced drug stability. Additionally, the nanofibers can mimic the extracellular collagen matrix, providing a suitable highly porous structural support for growing cells that facilitate and accelerate skin burns healing. This review gives an overview of the current state of electrospun fibers loaded with natural bioactive compounds as a biomedical system for skin burn treatment.

## 1. Introduction

Burns remain a major public health problem globally, resulting in increasing numbers of disabilities and preventable deaths each year. Burn trauma can be caused by heat (flame or scalding), freezing, electricity, chemicals, radiation, or friction [1]. Burned skin undergoes varied and complex processes after trauma, including coagulation, inflammation, cell proliferation, and tissue remodeling. These processes can be negatively affected due to invasion of infectious species, dehydration, and repeated additional trauma caused by surgery or wound cleaning [2].

In recent decades, efforts have been made to investigate a range of natural bioactive compounds (for example, plant extracts, oils of plant and animal origin, honeys, among many others) that have been traditionally used as curative agents, in addition to others that have been reported as having potential use in dermatology, particularly in the area of wound and burn healing [3]. The wound healing process can be achieved through the individual or combined action of bioactive agents and/or agents that promote angiogenesis, epithelialization, collagenation, or wound contraction. Natural products have a variety of components, including alkaloids, flavonoids, and terpenoids that are abundantly present in different sources of medicinal plants, marine life, fruits, and vegetables [4]. In the case of treatments for burns, the importance lies in the capacities they could possess both in the control of infections by common and resistant microorganisms and in the regeneration of the affected tissues, having in both cases low adverse effects. However, some of these bioactive compounds are chemically unstable, poorly soluble, susceptible to oxidative degradation, or have low bioavailability [5]. In this context, encapsulation is an alternative method that can be used to protect these compounds (polyphenols, vitamins, minerals, fatty acids, proteins, peptides, probiotics, etc.) [6], improve their physicochemical functions, and, in the application stage, improve their bioavailability by ensuring controlled release at specific sites [7,8]. Several encapsulation methods have been reported, such as nanoliposomes, nanoemulsions, coacervation, ionic gelation, electrostatic complexing, emulsification-evaporation of the solvent, nanoprecipitation, supercritical fluid, electrospray, etc. [9]. Electrospinning is a widely used, innovative encapsulation methodology. It is a simple and versatile technique for producing fibers loaded with bioactive compounds. The diameters of these fibers range from a few nanometers to micrometers, with controllable porosity, high contact surface area, length–diameter ratio, and specific surface area that improve cellular respiration, skin regeneration, and hemostasis, in addition to preventing infections, properties that allow potential applications in tissue engineering and drug delivery systems [10,11]. Therapeutic agents can be incorporated, and functionalized electrospun mats can be used as effective, stable wound dressings. The selection of polymers is fundamental in electrospinning, both in the spinning process, and later in the application to the fibers.

The electrospinning process requires that the material to be used, generally polymers, have a high molar mass, a critical mass concentration (to formulate viscous solutions), and a minimum electrical conductivity, in order to form fibers [12]. Factors such as the polymer concentration, viscosity, elasticity, polarity, and conductivity of the solution determine whether it can be electrospun into fibers, and also have an important effect on their morphology [13,14]. Polymers can be developed with a range of characteristics for different areas of application, with flexible or rigid fibers, air permeability, moisture permeability, controlled release of molecules, high biocompatibility, or adequate biodegradability; they can be made non-toxic, antimicrobial, and with enzyme activity or inhibition; and they can promote cell proliferation, resist high temperatures, or conduct electricity, etc. [15,16,17].

Traditionally, the burn healing process included the use of dressings with which the affected area was wrapped to protect it from friction and prevent infection. Now research points to the development of curative treatments involving devices that not only form a physical barrier, but also provide a variety of advanced features to enhance the healing process. The devices developed by electrospinning are economically viable and physicochemically promising in the burns treatment [18,19,20,21].

Electrospinning offers significant advantages over other existing wound and burn treatment, such as hydrogels or conventional dressings. Electrospun fibers have a structure very similar to the extracellular matrix (ECM) of the skin, which allows cells to migrate, adhere, and proliferate; this accelerates and facilitates the healing process [22,23]. In addition, electrospun fibers with antimicrobial properties can be developed, which could prevent complications derived from infections, a very common occurrence in this type of injury [19,24,25]. Despite these obvious advantages, electrospinning could also have some practical disadvantages or limitations, such as low cell infiltration and inadequate mechanical strength for load applications [23].

In general, the articles reviewed show that devices developed for burn treatment must possess important specific key functions such as: (1) preventing the invasion of microorganisms and reducing infection; (2) slowing dehydration and maintaining a favorable moist environment around the wound; and (3) mimicking the structural and biological characteristics of the ECM to promote complete regeneration of injured skin tissue [26].

The main of the present work was to analyze and categorize the physicochemical and structural properties of electrospun fibers with therapeutic potential for the treatment of skin burns, the polymers used, and the natural bioactive compounds encapsulated.

## 2. Literature Reviewed

The keywords used were “nanofibers”; “electrospun”; “skin” and “burns” in the Web of Science (WOS) database. Forty-nine publication results were found for 2006 to 2021. The type of bioactive compound contained in the electrospun fibers used in the different studies was analyzed, and two categories were established: the natural bioactive compounds used in 24 publications, which were finally selected for this review, and the synthetic bioactive and/or non-biological compounds (Table 1) used in 16 publications, which were not selected because they did not refer to the topic of this review. The subject matter of the remaining publications found does not coincide with that of this review, so they were not included either.

## 3. Categorization and Characterization of Electrospun Fibers

The electrospinning method consists of pumping a simple or complex polymer solution through a capillary subjected to a high-voltage electric field [13,27,32]. Due to electrostatic repulsions from the Taylor cone formed at the tip of the capillary, the polymer solution travels to a collector that may have an opposite electrical charge or be grounded [42,43] (Figure 1). The jet is stretched and churned as it travels to the collector, the solvent evaporates during this process, and a solid, non-woven, fibrous matrix is deposited on the collector [44]. The alignment of the fibers in the collector is random (Figure 2); however, methods such as rotational, magnetic, gap, or post-drawing are being studied to induce a more ordered alignment in order to expand the mechanical properties and improve a variety of physical properties [45]. Many parameters affect the electrospinning process, such as operational variables (electric field intensity, fluid flow, distance to the collector plate), properties of the solution (concentration, viscosity, electrical conductivity, voltage surface, dielectric properties), and environmental parameters such as humidity and temperature [13].

Analyzing the publications reviewed, it was found that—despite the differing conditions such as voltage, flow ratio and distance between ejector and collector—the fibers obtained presented diameters on the nanometer scale in all cases except in the studies of Kadakia et al. [47], Li et al. [22], and Ilomuanya et al. [20] (Table 2).

Another characteristic of electrospun fibers that must be identified is whether they are of the “blend-composite” or “core/shell” type, as these present differences in structure and in the controlled release of bioactive compounds (Figure 3).

Simple fibers: these are the most basic type of electrospun fiber. The fiber consists of a single type of material, which allows the bioactive compounds to be loaded due to its intrinsic characteristics. Only one publication [48] using this type of fiber was identified in this review.

Blend-composite fibers: It is important to establish the differences between blend fibers and composite fibers. The blend fibers are produced by mixing two or more materials, polymer–polymer or polymer–small molecule (drug, protein, antioxidant, etc.) until a homogeneous solution is formed, which is subsequently electrospun [49,50]. On the other hand, composite fibers can be developed by different types of mixtures, such as polymer–polymer (organic), polymer–inorganic, and inorganic–inorganic [51]. These types of fibers are characterized as having at least two different phases, which can be developed in situ, by film stacking or coating by rotation, or impregnation in solution [52,53]. In this review, we identified 21 publications that developed these types of fibers. This simple and effective method of preparing devices loaded with bioactive compounds for therapeutic applications such as burn treatment allows for the controlled release of the drug and has adequate physical-mechanical characteristics for such an application. A variant of the fiber blend is called “emulsion/fiber blend”, in which the materials are mixed with an emulsion containing the bioactive compound. Hajiali et al. [48] used this technique to encapsulate an essential oil that had been added to the oil/water (O/W) emulsion.

Core/shell fibers: Core/shell electrospinning, also called coaxial electrospinning, is a modification of conventional electrospinning, characterized by the use of sample ejection capillaries arranged for the injection of a solution into the other solution. The core/shell fibers have two clearly different sections, a central core formed by a solution, and a shell or outer layer formed by another solution [54]. Core and shell can encapsulate drugs independently [13,19,22]. The main advantage of this structure is that the shell polymer helps to protect the drug from direct exposure to the deleterious external biological and ambient environments. The core–shell structure is also useful for reducing the burst-release phenomenon. Fibers developed by this method allow controlled release in two different phases: the first occurs in the shell and can be used to treat acute inflammatory responses in primary wound healing because it occurs first; the second phase, release of the core compounds, occurs later, so it can be used to deliver compounds required in later stages of the burn healing process. This was the strategy proposed by Li et al. [22]: rapid release of antimicrobial peptides loaded in the shell, followed by controlled release of curcumin from the core. It was also used by Ramalingam et al. [19] to generate phased release of the drug minocycline hydrochloride and natural extract of *G. sylvestre*, thus obtaining a synergistic effect of infection control and accelerated burn healing.

## 4. Materials Used for Production of Electrospun Fibers

The electrospun fibers with biomedical applications for skin burn treatment observed in the publications reviewed were produced using either a single polymer type or a polymer blend. Some of these polymers, such as poly(vinyl alcohol), polyurethane, or gelatin, fulfil a purely structural function in the fiber, while others, such as chitosan, alginate, or poly(3-hydroxybutyrate-co-3-hydroxyvalerate), provide structural functions and/or bioactivity. The most used natural materials in the development of electrospun fibers were chitosan and gelatin, while the most widely used synthetic materials were poly(vinyl alcohol) and polycaprolactone. The polymers reviewed are described below (Table 3).

### 4.1. Natural Materials

#### 4.1.1. Chitosan (CH)

Chitosan are random copolymers of N-acetyl d-glucosamine and 2-amino-2-deoxy-β-d-glucosamine residues, achieved by deacetylation of chitin (linear polysaccharide mainly composed of β (1→4) units linked to N-acetyl-2-amino-2-deoxy-d-glucose residues) [68]. This substance was discovered in 1859 and is the next most abundant compound in nature after cellulose. It has valuable properties such as biocompatibility, biodegradability, hydrophilicity, non-toxicity, high bioavailability, simplicity of modification, favorable moisture permeability, excellent chemical resistance, ability to form films, gels, nanoparticles, microparticles, and beads, as well as affinity for metals, proteins, and colorants [69]. Chitosan was used by Bayat et al. [21] to develop electrospun nanofibers loaded with bromelain, evaluating the recovery of burned skin in an animal model as one of the responses. The study concluded that chitosan-bromelain nanofibers at 2% *w*/*v* have a higher wound healing activity than chitosan-only nanofibers in the animal model tested. This may be due to chitosan having an antimicrobial activity that prevents infection in the burn; however, it does not promote healing or stimulate earlier or faster regeneration of the tissues, activities attributed to bromelain [70]. Chitosan was also used by Antunes et al. [18] in the development of an electrospun membrane composed of arginine-modified deacetylated chitosan (CH-A) for use as a wound dressing. The results showed improved tissue regeneration and faster wound closure when the modified membranes were used, compared to the unmodified membranes. These studies illustrate that this polymer has good biocompatibility and positive effects on the healing of skin wounds caused by burns. Talukder et al. [57] developed three-layer electrospun fiber structures. The inner layer was composed of PVA/CH/Fibrin; CH was used to enhance antibacterial inhibition in wounded skin—which was demonstrated against *E. coli* and *S. aureus* in antibacterial activity measurement assays.

#### 4.1.2. Collagen (COL)

Collagen is the most abundant protein in the animal body, representing approximately 30% of the total. It is the main component of the ECM and is vital for the mechanical protection of tissues and organs, and the physiological regulation of the cellular environment; it is widely used for biomedical and pharmaceutical applications [66]. Collagen is a biodegradable, non-toxic protein, with higher biocompatibility than other natural polymers, and is only weakly antigenic; it is also a surfactant and can penetrate a lipid-free interface [71]. Collagen molecules are made up of three polypeptide chains. These chains, aligned in parallel and wound to the left in a polyproline type II (PPII) helix, wrap around each other to form a triple helix to the right that is stabilized by hydrogen bonds between chains and within the n chain → π * interactions [72]. Venugopal et al. [63] used collagen mixed with polycaprolactone (PCL) to develop an electrospun nanofiber membrane. The study concluded that the collagen nanofibrous membrane mixed with PCL promotes greater cell adhesion, proliferation, and dissemination of the dermal fibroblast for wound healing compared to PCL membranes only; this better cellular response to the COL/PCL membranes may be due to the increased porosity and improved mechanical properties that collagen provides in electrospun nanofiber membranes developed with this mixture. Sadeghi-Avalshahr et al. [64] developed electrospun PLGA nanofiber scaffolds to which they added collagen by two methods: the first was to make a PLGA/COL mixture that was later electrospun; in the second, collagen was added to the surface of the nanofibers after electrospinning, using chemical methods. The authors then compared the mechanical and biocompatibility properties of the scaffolds produced by the two methods. For the first method, they prepared a 20% (*w*/*v*) solution of PLGA/COL with a weight ratio of 4:1 in 1,1,1,3,3,3-hexafluoro-2-propanol (HFIP), obtaining smooth fibers with small pore size. These PLGA/COL nanofiber scaffolds showed lower mechanical properties than pure PLGA nanofibers and those obtained by the second method. The PLGA/COL scaffolds showed cytotoxicity against keratinocyte cells. When the degradation rate was evaluated, the PLGA/COL scaffolds degraded approximately five times faster than the controls and the scaffolds created by the second method. Ilomuanya et al. [20] used collagen to modify the mechanical properties of PLA. It should also be noted that the biochemical interaction between cells and collagen resulted from the binding of collagen I to cell membrane receptors, mediated by fibronectin, an ECM glycoprotein. The improved interaction between keratinocytes and scaffolds would invariably facilitate wound healing through fibronectin mediation, especially in situations where the healing process has been impaired.

#### 4.1.3. Gelatin (GE)

Gelatin is a heterogeneous mixture of peptides derived from collagen proteins obtained by procedures that involve the destruction of cross-links between polypeptide chains, together with some breaking of polypeptide bonds [73]. This polymer is widely used in activities related to the food, pharmaceutical, and cosmetic industries due to its excellent biocompatibility, easy biodegradability, and weak antigenicity; in addition, it is easily obtained by extraction from animal tissue such as skin, muscles, and bones [74]. Jin et al. [60] used gelatin as a biodegradable polymer mixed with PCL in the development of nanofibers that allowed fibroblast cell proliferation; they used this as a positive control in the study. Zhang et al. [16] used a mixture of gelatin with silk fibroin to develop nanofiber dressings loaded with Astragaloside IV, evaluating the therapeutic effects on wounds such as acute burn trauma. Gelatin was selected in this study due to its physical and mechanical properties, as it has high tensile strength, low ductility, and good air permeability; in addition, gelatin has a morphology similar to that of the dermis, as well as low antigenicity and rapid tissue degradation and absorption. Mayandi et al. [24] developed gelatin nanofibers loaded with ε-polylysine, which were cross-linked using polydopamine (pDa); this method was selected because the electrospun fibers of this polymer lack adequate mechanical stability and show a high degree of swelling, which limits their biomedical applications [75]. Talukder et al. [57] developed three-layer electrospun fiber structures using gelatin for the outer layer to absorb exudates from wounded or burned skin.

#### 4.1.4. Keratin

Keratin is a fibrous protein present in mammal hair, wool, quills, and horns (α-keratins) and in the feathers, claws, and beaks of birds and reptiles (β-keratins). Due to its abundance in nature and its ability to enhance cell proliferation, it is an ideal material for a variety of biomedical applications, ranging from scaffolds for cell growth to drug delivery. Given keratin’s rather poor mechanical properties and its low molecular weight (ranging from 10 to 60 kDa), it is often combined with synthetic polymers, such as PEO, PCL, or PVA, which serve as adjuvants to improve processability. Kossyvaki et al. [59] showed that keratin has high antioxidant activity (uptake of the 2, 2-diphenyl-1-picryl-hydrazyl-hydrate free-radical (DPPH)), which has been scarcely reported in the literature. The antimicrobial activity determination tests showed that the PVP/keratin fibers had a bactericidal effect limited to *S. aureus*. The authors also established that keratin is not cytotoxic, nor does it inhibit the growth and proliferation of primary human dermal fibroblasts (α-HDF cells). Finally, an important finding was that the keratin-based patches with and without cinnamon essential oil were able to reduce in vivo the expression of pro-inflammatory factors (IL-1b y IL-6) by 5–7 fold.

#### 4.1.5. Poly(3-Hydroxybutyrate-co-3-Hydroxyvalerate) (PHBV)

Poly(3-hydroxybutyrate-co-3-hydroxyvalerate) is a natural polymer with thermoplastic properties that can be produced by bacterial fermentation, but the process is not economically competitive with the production of polymers from petrochemical products [76]. PHBV is a biocompatible and biodegradable polymer investigated for various tissue engineering applications. Sundaramurthi et al. [26] studied the adhesion, proliferation, and epidermal differentiation of mesenchymal stem cells (BM-MSCs) in PHBV nanofibers. The results obtained demonstrated that this polymer provides a medium permitting the cellular differentiation of BM-MSCs; it can be used in a device based on electrospun nanofibers as immediate cover for third degree burns, traumatic ulcers, and diabetic wounds.

#### 4.1.6. Silk Fibroin (SF)

Silk fibroin is a protein obtained from the cocoons of the *Bombyx mori* worm; up to 90% of its composition consists of the amino acid glycine, alanine, and serine [67]. SF possesses good biocompatibility, elasticity, toughness, as well as suitable mechanical properties and biodegradability with controllable degradation rates [77]. Zhang et al. [16] developed electrospun nanofibers using two natural polymers, SF and gelatin. The authors explained that SF was selected due to its good biocompatibility and because it does not release products that irritate the skin or have toxic side effects, despite the fact that its biodegradation is partial and slow. SF has good, sustained release performance, high flexibility, and good air permeability and moisture permeability. The disadvantages of SF are compensated by the properties that gelatin gives to the polymer mixture. Thus, the scaffold composed of silk fibroin and gelatin should present structural and chemical similarities with the environment of the ECM, making it a promising device for the therapeutic treatment of skin burns. The results obtained reinforce this idea: the SF/gelatin nanofibers loaded with Astragaloside IV not only promoted wound healing, but also inhibited scar complications.

#### 4.1.7. Sodium Alginate (SA)

Sodium alginate is a natural polysaccharide derived from brown seaweed, used as a gelling agent and for spherification. It is the monovalent salt form of alginic acid, comprising 1,4-linked β-d-mannuronic (M), and α-1-guluronic (G) acid units [78]. Sodium alginate was used by Hajiali et al. [48] to create nanofibers containing lavender (*Lavandula angustifolia*) essential oil to develop nanofiber dressings with antibacterial activity against *S. aureus*, and to inhibit the production of pro-inflammatory cytokines in the area of the burn. The authors specified that the appeal of sodium alginate is its ability to release bioactive compounds and maintain a moist environment around the wound, promoting tissue granulation and re-epithelialization. The electrospun dressings had an active effect on promoting burn healing; the strong anti-inflammatory action of the sodium alginate was evident in all the tests carried out. Talukder et al. [57] developed three-layer electrospun fiber structures, in which the middle layer was produced with PVA/SA. SA was used to enhance antibacterial inhibition in wounded skin.

### 4.2. Synthetic Materials

#### 4.2.1. Poly(Ethylene Glycol) (PEG)

Poly(ethylene glycol) is the most commonly applied non-ionic hydrophilic polymer in the field of polymer-based drug delivery [79]. PEG is also known as poly(ethylene oxide) (PEO) or polyoxyethylene (POE) depending on its molecular weight. The structure of PEG is commonly expressed as H−(O−CH2−CH2)n−OH [80]. PEGs reduce the tendency of the particles to aggregate by steric stabilization, thus, producing formulations with greater stability during storage and application [81]. Li et al. [22] developed a core/shell nanofiber membrane using a mixture of PEG/PLA polymers, later mixed with curcumin for the core of the nanofiber. The purpose of using PEG/PLA was to contain the curcumin and generate a slower controlled release than that obtained with the fiber shell components. These effects have a positive impact on the therapeutic performance of the dressing. The authors do not explain the reason for the use of PEG, but it could be due to its favorable viscoelastic properties and its non-toxicity.

#### 4.2.2. Poly(Lactic-co-Glycolic Acids) (PLGA)

PLGA is a synthetic polymer from the polyester family. Polyesters have been widely studied as delivery vehicles for drugs, proteins, and various other macromolecules such as DNA, RNA, and peptides. This polymer presents many advantages, most especially its biocompatibility. Other benefits include its adequate rate of biodegradation and the production of non-toxic biodegradation products; the approval by the Food and Drug Administration (FDA) of more than 20 PLGA-based pharmaceutical products to date [82]; and the potential to modify surface properties to provide better interaction with biological materials [83]. Sadeghi-Avalshahr et al. [64] used PLGA to produce electrospun nanofibers, which are characterized by high hydrophobicity. This could negatively affect the efficiency of cell growth and adhesion. The authors mixed PLGA with collagen using two methods, as described in Section 4.1.2, since this natural polymer would help make the fibers less hydrophobic. The authors analyzed and compared the mechanical and biocompatibility properties of nanofibers obtained by these two methods, concluding that the samples obtained with the second method were more suitable for applications as skin substitutes.

#### 4.2.3. Poly(L-Lactic Acid) (PLLA)

PLLA is a biodegradable polymer widely used in bioengineering for its biocompatibility, and also as a drug delivery vehicle, although characteristics such as high rigidity and hydrophobicity limit its use in some areas [84]. Li et al. [62] developed PLLA-based electrospun nanofibers by incorporating functional polyhedral oligomeric silsesquioxane (POSS) nanoparticles, which are known for their organic–inorganic structure and biocompatibility. This nanofiber system was used to encapsulate plasmid DNA encoding angiopoietin-1 (pAng). PLLA/POSS nanocomposite can form a porous fiber structure and shows greatly improved mechanical performance. The results showed that a pAng-loaded PLLA/POSS scaffold effectively promoted angiogenesis and dermal wound healing.

#### 4.2.4. Poly(Vinyl Alcohol) (PVA)

PVA is a water-soluble synthetic polymer with excellent film-forming, emulsifying and adhesive properties [85], specifically for various pharmaceutical and biomedical applications. PVA has a relatively simple chemical structure with a pendant hydroxyl group. The monomer, vinyl alcohol, does not exist in a stable form, but rearranges to its tautomer, acetaldehyde [86]. Saeed et al. [65] developed a three-layer nanofiber dressing using electrospun-based PCL and PVA for the treatment of wounds and burns. The intermediate layer of the three was PVA; its function was to absorb the exudates produced by the damaged tissue area. The results of the absorbency test revealed that the addition of a layer of PVA increases the absorbency by a factor of three; this points to the effectiveness of this dressing layer in absorbing exudates. Elshishiny and Mamdouh [25] fabricated a scaffold consisting of an upper electrospun chitosan-poly(vinyl alcohol) layer and a lower synthetic polymer xerogel layer. The two layers are fixed together using fibrin glue as a middle layer. The authors selected PVA to facilitate the electrospinning process, and also considering its various significant and promising characteristics, including high degrees of swellability and elasticity, rubber-like structure, bioadhesiveness, non-carcinogenicity, and ease of handling.

#### 4.2.5. Poly(Vinyl Pyrrolidone) (PVP)

Poly(vinyl pyrrolidone), also commonly called polyvidone or povidone, is a water-soluble polymer made from the monomer N-vinylpyrrolidone [87]. PVP with molecular weights (Mw) from 2500 to about 1 million is mainly obtained by radical polymerization in solution. Some of the most important applications of PVP in the pharmaceutical field are as binding agents or film-forming agents for tablets, and as solubilizing agents for injections [87]. Li et al. [70] developed core/shell nanofibers in which a mixture of PLA/PVP with antimicrobial peptides (AMPs, HHC36) was used as the shell layer. They were able to optimize the shell’s PLA/PVP ratio, achieving increased thermal stability, mechanical properties, and swelling capacity.

#### 4.2.6. Polycaprolactone (PCL)

Polycaprolactone is a hydrolytically degradable polymer of the aliphatic polyester type. The rate of hydrolysis of this polymer increases with time, which has been attributed to the high reactivity of the terminal ester and the kinetics of autocatalysis [88]. PCL is compatible with soft and hard tissue, including resorbable suture, and is used as a drug delivery system; it has recently been used to develop a bone graft substitute [63]. It was also the most widely used polymer in the literature reviewed. Jin et al. [60] developed electrospun PCL nanofibers to encapsulate individual plant extracts of *Indigofera aspalathoides* (*I. aspalathoides*), *Azadirachta indica* (*A. indica*), *Memecylon edule* (*M. edule*), and *Myristica andamanica* (*M. andamanica*). PCL provided a suitable environment for fibroblast cell differentiation, with PCL/*M. edule* being the most suitable substrate for skin tissue engineering. Venugopal et al. [63] developed a PCL nanofibrous membrane mixed with collagen that exhibited cell adhesion and proliferation, and dissemination of dermal fibroblasts for wound healing. This device could have a potential for application in tissue engineering as a dermal substitute in the treatment of skin defects and burns. The role of PCL in the PCL/PVA/PCL multilayer devices reported by Saeed et al. [65] was to load the curcumin and to provide the mechanical properties for a controlled release of curcumin. The PCL/PVA/PCL multilayer devices achieve excellent benefits, being soft, conformable, and non-adhesive, and offering an easy solution for covering irregular-shaped wounds, as well as anti-bacterial characteristics. Ramalingam et al. [19] developed core/shell fibers using PCL as the main component of the shell, where they encapsulated a synthetic drug. When they performed the release tests, the release of natural extract encapsulated in the core of the fibers was negligible. To reverse this behavior, the authors added GE to the shell composition. An explosive release of core components was observed when the fiber shell was composed of the PCL/GE blend; the authors attributed this change in release kinetics to the greater wettability of the PCL/GE nanofibers.

#### 4.2.7. Poly-D,L-Lactic Acid (PDLLA)

Poly-D, L-lactic acid is an amorphous polymer randomly composed of repeating units of L-lactic acid and D-lactic acid. Steffens et al. [61] incorporated Spirulina microalgae into electrospun nanofiber scaffolds of PDLLA polymer for the treatment of skin wounds caused by burns. Macroscopic analysis of the scaffold groups showed better healing compared to the control group, although the findings were not completely conclusive.

#### 4.2.8. Polylactide (PLA)

Polylactide or polylactic acid is a biodegradable thermoplastic polyester derived from sources such as corn starch, cassava starch, and sugar cane. PLA is only soluble in a narrow range of solvents such as tetrahydrofuran, dioxane, chlorinated solvents and heated benzene, which is a significant limitation on its use [89]. Li et al. [22] used PLA in the development of core/shell nanofibers in both the core and the shell. A mixture of PLA/PVP with antimicrobial peptides (AMPs, HHC36) was used for the shell layer, while PLA/PEG blended with curcumin was used as the core. This device presented controlled release from both the shell and the core, in addition to mechanical characteristics and porosity that favored interaction with the burn site. Ilomuanya et al. [20] selected PLA as it is a biocompatible, non-cytotoxic polymer widely used in biomedical applications; however, when used individually it generates highly rigid, hydrophobic fibers. To improve its mechanical properties, it must be mixed with another polymer, in this case the authors used collagen.

#### 4.2.9. Polyurethane (PU)

Polyurethane is a polymer composed of organic units linked by carbamate (urethane) bonds. It is a versatile material with great potential due to its specific mechanical, physical, biological, and chemical properties [90]. Its hardness, durability, biocompatibility, and degradation rates can be adapted according to the application [91]. Kim et al. [2] developed electrospun nanofiber-based dressings for the therapeutic treatment of burns using PU as a base polymer, badger (*Meles meles*) oil as a healing agent, and silver nanoparticles (AgNP) as an antibacterial agent. The authors obtained smooth, uniform-diameter, hydrophobic nanofibers, although this last characteristic was modified with the addition of oil and silver nanoparticles, which improved cell adhesion during wound healing.

## 5. Natural Bioactive Compounds Used in Electrospun Fibers

The articles reviewed show a great variety of natural bioactive compounds encapsulated in electrospun fibers (Table 4); their biological effects can be classified either by their antimicrobial effects or by their acceleration of the wound healing process.

### 5.1. Natural Antimicrobial Bioactive Compounds

#### 5.1.1. Badger (*Meles Meles*) Oil

The badger (*Meles meles*), belonging to the *Mustelidae* family, is a highly adaptable, medium-sized carnivore/omnivore that occupies a variety of biomes throughout temperate Eurasia [92]. In traditional Chinese and Korean medicine, badger oil was used to treat wounds and burns [93]. The major fatty acid component is oleic acid (30.43%), with palmitic acid (20.51%), linoleic acid (10.62%), stearic acid (8.82%), and palmitoleic acid (7.04%) also present [2]. Kim et al. [2] developed electrospun nanofibers based on PU to which they added badger oil, or badger oil/Ag nanoparticles (AgNPs). Pure badger oil cannot produce continuous, uniform nanofibers so it must be mixed with a synthetic polymer, in this case PU, to provide it with electrospinning capability. The presence of badger oil allowed the generation of a series of connections formed between the fibers, which did not occur in fibers that only contained PU; this improved cross-linking in the electrospun matrix. Furthermore, fibers were obtained from the PU/badger oil mixture with average diameter approximately 93 nm smaller than that of PU fibers (520 nm → 427 nm). Despite the fact that badger oil is rich in oleic acid, which has antibacterial properties [94], PU/badger oil fibers showed no inhibition of *E. coli* (Gram-negative) or *S. aureus* (Gram-positive); this inhibitory effect is shown by PU/badger oil/Ag fibers, showing that AgNPs play an important role as antibacterial agents. The addition of badger oil to the PU resulted in a reduction of the contact angle to about 120.0 ± 0.14°, and the PU/badger oil/Ag mats became hydrophilic. In this study, the contribution of badger oil was rather complementary within the PU/badger oil/Ag set. The important antibacterial effect, and the improvement in the cytocompatibility of the electrospun fibers, was attributable to the addition of AgNPs.

#### 5.1.2. Olive (*Olea Europaea* L.) Oil

Olive (*Olea europaea* L; *Family- Oleaceae*) oil is rich in essential vitamins, monoenoic and dienoic fatty acids, and other natural nutrients. It is rich in oleic acid (monounsaturated fatty acid), which makes it susceptible to oxidative rancidity. Amina et al. [16] developed a scaffold based on electrospun nanofibers of PU mixed with Ag nanoparticles, in which they encapsulated olive oil (5–10%). Solutions were prepared in N,N-dimethylformamide/tetrahydrofuran (1:1 *v*/*v*) medium. Scanning electron microscopy (SEM) showed the oil droplets encapsulated within nanofibers, which presented local broadening as a result. The inhibition studies of *E. coli* and *S. aureus* showed that the PU/olive oil nanofibers have antimicrobial activity; however, this is lower than that presented by the PU/olive oil/Ag nanofibers. This is easily explained by the known antimicrobial activity of Ag, which caused total inhibition of bacterial colonies. The fibers showed null cytotoxicity, so the authors claim that olive oil has potential use in the development of scaffolds for the treatment of wounds and burns.

#### 5.1.3. Chitosan (CH) & CH/l-Arginine (CH-Arg)

Chitosan is a copolymer widely used in electrospinning since it is biocompatible, biodegradable, and described as having antimicrobial, healing, anti-inflammatory, and homeostatic capacity [18]. Despite this, CH has a very high viscosity that limits its versatility in electrospinning [95]; however, this limitation is overcome by dissolution with strong acetic acid and strong trifluoroacetic acid or mixing with other polymers such as PEG or PVA [96]. These bioactivities make CH an ideal candidate for the development of devices based on electrospun nanofibers for the treatment of burns and wounds in the skin and tissues. Antunes et al. [18] mention that its antimicrobial activity may be the result of the interaction of positively charged amino groups with negatively charged groups present on the surface of bacterial cells. They also state that this bioactivity can be increased by adding positively charged amino acids such as l-arginine. Researchers developed CH-D/CH-Arg (60:40 *w*/*w*) soft nanofibrous membrane at a final concentration of 8% (*w*/*v*), using 2, 2, 2-trifluoroacetamide (TFA) and dimethyl carbonate (DCM) (70:30 *v*/*v*) as solvent. The membranes showed total inhibition of the bacterial strains *S. aureus* and *E. coli.* This demonstrates the efficacy of antimicrobial control when l-arginine residues are coupled to the CH polymer. The results of the cytotoxicity and cell proliferation tests indicate that l-arginine coupling does not affect the biocompatibility of the membranes. In vivo tests demonstrate that the membranes containing CH-Arg have a healing effect on the burn compared to control membranes without the conjugation of l-arginine to CH, or with the natural healing process alone. Elshishiny and Mamdouh [25] developed three-layer nanofiber scaffolds, in which the upper layer was made up of PVA/CH (3%). This layer expressed antibacterial properties against *S. aureus* and *E. coli*. The authors also described that the degree of CH deacetylation in some cases could affect cell migration, as observed in some trials in this study. Ketabchi et al. [56] developed electrospun fibers of CH/PEO-actinidin; when healing tests were carried out in an animal model, the CH/PEO nanofibroses without the enzyme showed a similar but slower recovery trend than when actinidin was present. No infection was observed in this study group, which could be attributed to the action of CH on possible infectious agents, but there were some hemorrhages after the ninth day.

#### 5.1.4. Curcumin (CU)

Curcumin is a yellow dye and active constituent of turmeric (*Curcuma longa*), an herbaceous plant of the *Zingiberaceae* family native to Southeast Asia. For centuries, turmeric has been used to treat various diseases in the traditional and herbal medicine of South and Southeast Asia [97]. Curcumin has antioxidant, anti-inflammatory, and anti-infective properties; however, it is extremely unstable in vivo with very low bioavailability, so encapsulation is a promising strategy. Saeed et al. [65] developed dressings with three layers of different curcumin polymers. The layers were generated with PCL and PVA polymers, forming lower and upper layers with PCL and an intermediate layer with PVA, all loaded with curcumin, in the order: PCL/CU; PVA/CU; PCL/CU. The antibacterial properties of the dressings were evaluated on *E. coli* and *S. Aureus*. Dressings with less than 8.5% curcumin do not reveal significant antibacterial properties; however, with ≥16% curcumin loaded in the dressings, 100% death of bacterial colonies occurs. When the cytotoxicity test was carried out, it was evidenced that dressings with 16% curcumin killed 40% of the cells, so they are less biocompatible with fibroblasts.

#### 5.1.5. Antimicrobial Peptides (AMP) HHC36

HHC36 are short natural peptides that have a broad spectrum of antimicrobial activity, limited contact time to induce death, and low susceptibility to the development of bacterial resistance [98]. Li et al. [22] created a core/shell nanofiber membrane to which PLA beads were subsequently added by electrospray. HHC36 was loaded in the shell (PLA/PVP), while curcumin was loaded in the core (PLA/PEG). The antimicrobial activity of the nanofiber membranes was tested in vitro against *E. coli* and *S. aureus*. The membranes were not cytotoxic; on the contrary, adhesion and proliferation of NIH/3T3 cells were observed on the surface. The charging efficiency of HHC36 was 70.6 ± 3.3%. The release of HHC36 from the shell was faster than release of CU from the core. This device could be used for the treatment of acute bacterial infection. Overall, the combined antibacterial activity and profound wound healing effects of this device make it an interesting option for treating skin burns.

#### 5.1.6. Manuka Honey (MH)

Manuka Honey is produced by endemic New Zealand bees, which pollinate the Manuka tea tree (*Leptospermum scoparium*) [99]. MH has an inhibitory behavior against aerobic, anaerobic, Gram-positive, and Gram-negative bacteria due to the enzymatic production of hydrogen peroxide, high osmolarity and acid pH; however, it has an alternative antimicrobial activity not related to hydrogen peroxide, but to methylglyoxal [47,99,100]. Although MH possesses interesting attributes for wound and burn healing, Kadakia et al. [47] evaluated its hygroscopic capacity for water absorption and retention in electrospun nanofibers, comparing them with those of the poloxamer 407 copolymer (P407). While MH has many positive attributes for wound healing, this study focused primarily on its hygroscopic abilities. Solutions of MH in 1,1,1,3,3,3-hexafluoro-2-propanol (HFP) were prepared, to which SF was added until a 10% (*w*/*v*) polymer solution was obtained. The nanofiber scaffolds featured relatively open pores with randomly oriented nanofibers. The cell proliferation assay showed that the MH scaffolds had a higher number of cells, which could be caused by increased release of growth factors from adult human dermal fibroblasts (hDFs) due to contact with MH. This study established that P407 scaffolds increased the wettability of the fiber surface, and that MH modulated moisture retention.

#### 5.1.7. ε-Polylysine (εPL)

This is a natural antimicrobial peptide produced from aerobic bacterial fermentation by *Streptomyces albulus*, applied as a safe food preservative against Gram-positive and Gram-negative bacteria; it may also be a promising antifungal agent [101,102]. Mayandi et al. [24] loaded εPL into electrospun fibers derived from GE; trials demonstrated that these peptides possess highly effective antimicrobial activity against antibiotic-resistant Gram-negative strains. The broad-spectrum antimicrobial properties demonstrated by εPL, together with its membranolytic action and the fact that it does not induce acquired resistance in *P. aeruginosa* or VRE (vancomycin-resistant *Enterococcus faecium*) strains, establish the potent antimicrobial properties of this cationic peptide.

#### 5.1.8. Gymnema Sylvestre Extract

*Gymnema sylvestre* (Apocynaceae) is a plant present in many regions of Asia, Africa, and Australia; it is widely used in traditional medicine for different purposes. *G. sylvestre* has antioxidant, antibiotic, anti-inflammatory, antiviral, gastro and hepatoprotective, anticancer and lipid-lowering activities, as well as hypoglycemic activity, the last point due to the presence of phytochemicals such as gurmarin, gymnemic acid, and gymnemasaponins [103]. Ramalingam et al. [19] developed core/shell fibers in which they encapsulated *G. sylvestre* extract in the core, and the synthetic drug minocycline in the shell. The authors evidenced the existence of synergy between the components of the plant extract and the drug in the control of Gram-positive and Gram-negative bacteria (*S. aureus*, *S. aureus* resistant to methicillin, *S. epidermidis*, *P. aeruginosa* and *E. coli*). The marked synergism between plant extracts and minocycline towards Gram-positive cocci is attributed to greater penetration of the combined products into the cell wall. The analyses showed that the extracts contained gymnemagenin, beta-sitosterol, lupeol and quercetin as major components. This work also reports in vivo evaluation of wound healing in a skin burn wound in a pig model; the results showed clear signs of reorganization of the connective tissue with a profusion of blood vessels after 32 days when the wound was treated with fibers loaded with *G. sylvestre* extracts and minocycline.

### 5.2. Natural Bioactive Compounds Producing Accelerated Burn Wound Healing

#### 5.2.1. Bromelain (Br)

Bromelain is a mixture of proteolytic enzymes present in all tissues of pineapple (*Ananas comosus*), with a similar function to papain and ficin. Its enzymatic activity depends on the thiol group of a cysteine residue in its active site [104]. One of the important pharmaceutical applications of bromelain is the enzymatic debridement of necrotic tissues from ulcers and burns [105]. Bromelain has been shown to interact with a variety of effectors and pathways involved in physiological processes such as inflammation, immune response, and coagulation [106]. Bayat et al. [21] developed chitosan/polyethylene oxide (CH/PEO) nanofibers loaded with bromelain (2% and 4% *w*/*v*) by the electrospinning method. Fourier transform infrared (FTIR) spectra showed the formation of possible intermolecular bonds between chitosan and bromelain in the nanofibers. The bromelain loading efficiency was 91% and 96% respectively in the two formulations: CH-2% and 4% *w/v* of Br. CH-2% *w/v* Br showed no cytotoxicity, whereas cross-linked CH-4% *w/v* Br showed significant cytotoxicity; this could be caused by glutaraldehyde vapor residues (cross-linking) in the fibers. Ch-2% *w/v* Br significantly decreased the burned area in in vivo tests in an animal model, achieving almost complete regeneration of the skin and hair. Br is known as an important natural debriding agent, and it is described that bromelain proteases can hydrolyze fibrin clots present in the burned area. In addition, it has the ability to enzymatically digest damaged areas of the ECM that contain collagen, elastin and laminin, accelerating the natural healing process of these lesions.

#### 5.2.2. Microalga Spirulina (*Arthrospira* sp.)

Spirulina is a microscopic, filamentous cyanobacteria (blue-green algae) generally produced from *Arthrospira platensis* and *Arthrospira maxima*. Spirulina contains significant amounts of vitamin B12 and provitamin A (β-carotene), minerals, carotenoids, and phycocyanins [107]. It also contains polysaccharides with anti-inflammatory effects and fatty acids with antibacterial and antifungal effects. These characteristics make it an ideal candidate for the development of nanofiber scaffolds for the treatment of burns and skin wounds [108]. Steffens et al. [61] developed a nanofiber scaffold using the polymer PDLLA 8% (*w*/*w*) and Spirulina 2% (*w*/*w*) in 1,1,1,3,3,3-hexafluoro-2-propanol (HFIP). Smooth nanofibers were obtained, without defects and with interconnected pores. The scaffolds showed the growth and proliferation of mesenchymal stem cells in their structure. PDLLA/Spirulina scaffolds showed more moldability and adherence to the wound compared with PDLLA in an animal model. However, no stimulus in regeneration of damaged tissues was observed after injury in these animals. The authors argue that this lack of accelerated regeneration may be due to the short analysis period (seven days), since they only observed the inflammatory phase; they, therefore, conclude that it is necessary to carry out studies with longer follow-up periods to evaluate the three phases of healing: inflammation, proliferation, and remodeling.

#### 5.2.3. Plasmid DNA Encoding Angiopoietin-1 (pAng)

The interaction between cells and their microenvironment is determined, among other things, by growth factors [109]. The growth, proliferation, migration, and formation of blood vessels from vascular cells is stimulated by angiogenic growth factors such as angiopoietin (Ang), vascular endothelial growth factor (VEGF), platelet-derived growth factor (PDGF), and endothelial growth factor (EGF) [62]. Li et al. [62] used the electrospinning technique to develop PLLA/POSS nanofibers capable of sustained release of pAng particles (pAng/TMC complex), significantly increasing the efficiency of gene transfection. The encapsulation efficiency of pAng in PLLA/POSS fibers was 89.5 ± 2.9%. For PLLA/POSS fibers, there was a linear sustained release of pAng over the 35 days tested. The efficiency of gene transfection was evaluated, and it was observed that the amount of Ang expressed in the PLLA/POSS/pAng fibers was 1.2 times higher than in the control group. pAng appears to promote cell proliferation in the nanofiber scaffold. Based on an in vivo evaluation, encapsulated pAng nanofibers not only stimulate angiogenesis, but also accelerate the rate of wound healing.

#### 5.2.4. Astragaloside IV

Astragaloside IV (AS-IV) is a purified natural active component of Astragalus membranaceus, widely used in traditional Chinese medicine [110]. Antioxidant, immunostimulant, and anti-inflammatory properties have been attributed to this plant. Due to its medicinal use, it is postulated that AS-IV could promote wound healing, neovascularization and keratinocyte migration and proliferation, and also prevent scar formation by regulating collagen expression. Zhang et al. [16] developed an SF/GT nanofiber dressing loaded with AS-IV and evaluated its effects in vivo in an acute trauma model. The nanofibers were prepared with a mixture SF and GT (25/75%) at 4% (*w*/*w*) using hexafluoroisopropanol as solvent, which was added to variable amounts of AS-IV dissolved in hexafluoroisopropanol. The SF/GT/AS-IV nanofibers were biocompatible with keratinocytes, and significantly faster wound healing was found than in the control group in the early stage of trauma. Histological tests showed that SF/GT/AS-IV promoted the formation of new blood vessels. Additionally, evidence was found that SF/GT/AS-IV could inhibit scar complications.

#### 5.2.5. α-Lactalbumin (ALA)

α-lactalbumin (ALA) is a whey protein (14.2 kDa) that is a constituent of mammalian breast milk. The main biological role of ALA is to regulate lactose biosynthesis in the mammary glands, facilitating milk production. It is an important source of bioactive peptides and essential amino acids, including tryptophan, lysine, and amino acids. Tryptophan is important for the synthesis of the neurotransmitter serotonin [111]. According to recent studies, serotonin could promote the healing of skin wounds in burn patients by improving the proliferation and migration of cells such as keratinocytes and fibroblasts [112]. Guo et al. [58] selected ALA as a rich source of tryptophan, a precursor of the neurotransmitter serotonin, to promote burn healing and reduce scar formation. The authors developed PCL/ALA electrospun fibers and evaluated the physicochemical attributes and burn healing efficiency in an animal model. The PCL/ALA fibers were not cytotoxic and also favored the proliferation and growth of fibroblasts after 24 h of incubation. This effect was dependent on the ALA concentration in the fibers, increasing as the ALA concentration increased. The authors also observed that PCL/ALA fibers accelerated the healing process in in vivo tests. ALA was observed to promote the proliferation and migration of neonatal fibroblasts and keratinocytes to the burn site, facilitating and accelerating the natural healing of tissues. The authors also highlight that PCL/ALA fibers accelerated the wound healing process by increasing the synthesis of collagen I, increasing the ratio of collagen I to collagen III and reducing the expression of α-SMA, consequently limiting scar complications.

#### 5.2.6. Fibrin

Fibrin is a polypeptide consisting of components of fibrinogen and thrombin that are active in blood clotting. It is essential for hemostasis and is an important factor in thrombosis, wound healing, and various other biological functions and pathological conditions [113]. Talukder et al. [57] conducted experiments to determine the optimum concentrations of the components of the electrospun fibers established using fibrin 1%. Electrospun PVA/CH/Fibrin fibers impacted skin regeneration, providing biocompatible traits, while fibrin functioned to stimulate proper blood clotting.

#### 5.2.7. Actinidin

Actinidin is a protease enzyme of the thiol protease type, with a molecular weight of 26,000, that acts in the same way as collagenases. It is abundant in kiwifruit (*Actinidia delicious*). Studies have shown that actinidin does not have the ability to hydrolyze the natural collagen molecule, but it can hydrolyze atelocollagen, which is partially digested collagen [114]. Ketabchi et al. [56] investigated the development of electrospun CH/PEO fibers to obtain optimized conditions. Actinidin was subsequently immobilized on the optimized CH/PEO nanofibrous patches by the 1-ethyl-3-(3-dimethylaminopropyl) carbodiimide/N-hydroxysuccinimide (EDC/NHS) activation procedure using fetal bovine serum (PBS) buffer (pH = 7). Based on the results obtained, a 20% actinidin enzyme concentration was chosen for immobilization on the surface of electrospun nanofiber patches. The actinidin enzyme, in addition to not being cytotoxic, induces cells to greater viability and proliferation. In the in vivo burn tests carried out on an animal model, the CH/PEO fiber patches with actinidin caused debridement and digestion of the necrotic tissue; moreover, there was no infection or bleeding in the affected area. In general, the nanofibrous patch with the enzyme actinidin played a role in the rapid process of angiogenesis, epithelialization, and collagen production.

#### 5.2.8. Quercetin/Rutin

Quercetin is a bioactive flavonoid hydrolyzed from rutin that is widely distributed in fruits and vegetables and is known for its powerful antioxidant activity. It has been used widely in botanical medicine and traditional Chinese medicine [115]. Quercetin and rutin have attracted increasing attention due to their potential for pharmacological use, including antioxidant, anti-inflammatory, and antimicrobial activities, protection of blood vessels and nerves, and pain management; they may also be able to promote wound healing. Zhou et al. [55] developed a membrane of electrospun fibers of PCL, chitosan oligosaccharides and quercetin/rutin, which has the potential to eliminate reactive oxygen species produced by inflammatory reactions; the antioxidant and antibacterial activities of quercetin/rutin may prevent wound healing being retarded by infections. The analysis of the trials showed that quercetin/rutin did not lose their antioxidant capacity when encapsulated in the electrospun fibers. In the antibacterial activity measurement tests, there was no difference between the membranes evaluated in the inhibition of *E. coli* at 12 h or 24 h, which indicates that the low solubility of quercetin compared to rutin was not favorable for the inhibition of *E. coli*; however, after 24 h the bactericidal activity was complete. In contrast, all electrospun membranes demonstrated *S. aureus* inhibition at 12 h. In general, PCL/CH-quercetin/rutin fibers presented excellent antibacterial activity and have the potential for use in burn dressings in the future.

### 5.3. Natural Bioactive Compounds with Antimicrobial Effects for Burn Wound Healing

#### 5.3.1. Lavender Essential Oil (*Lavandula Angustifolia*)

Lavender (*Lavandula angustifolia* Mill.) is one of the most widely cultivated essential oil crops in the world. It has been used as an anxiolytic drug, mood stabilizer, sedative, spasmolytic, antihypertensive, antimicrobial, analgesic, and accelerator of wound healing [116]. Lavender essential oil is said to possess antibacterial, antifungal, carminative (smooth muscle relaxant), sedative, antidepressant properties, and effective activity for healing burns and insect bites [117]. Lavender oil is extracted commercially by steam distillation. Hajiali et al. [48] prepared solutions for electrospinning by dissolving SA and PEO in distilled water and later in DMF, adding the surfactant Pluronic F127, and finally adding 5% *v*/*v* lavender oil to emulsify the oil in the aqueous phase before electrospinning. The authors studied the in vitro release of linalool, caryophyllene, and caryophyllene oxide, characteristic components of lavender oil. The fibers efficiently released the oil, peaking after 6 h. The SA-PEO/lavender oil fibers presented inhibition of bacterial colonies of *S. aereus*; this is due to the presence in the lavender oil of linalool, and also terpinen-4-ol, both of which present antimicrobial activity. SA-PEO/lavender oil also presented the most important in vitro anti-inflammatory activity, achieving a decrease in IL-6 (66%) and IL-8 (49%) concentrations. The authors also evaluated the activity of the SA-PEO/lavender oil dressings in vivo in an animal model, showing effective control of cytokine production and inflammation, and enhanced healing that does not leave marks or spots on the skin.

#### 5.3.2. Cinnamon (*Cinnamomum verum*) Essential Oil

Cinnamon oil can be obtained from the leaves or bark of different types of cinnamon plants. The essential oil extracted from the bark of *Cinnamomum verum* exhibits a wide variety of therapeutic effects, including antimicrobial activity and antioxidant properties that can actively affect skin inflammation. Studies show that it is rich in the aromatic compounds cinnamaldehyde and eugenol, as well as phenols and flavonoids [118]. Kossyvaki et al. [59] encapsulated cinnamon oil in PVP/keratin electrospun fibers. In the tests to determine antioxidant activity, they showed that higher concentrations of cinnamon oil in the fibers cause a greater decrease in DPPH free radicals, the maximum limit being 10% cinnamon oil. After this concentration, no antioxidant activity occurs. Evidence was reported that cinnamon oil had an antibacterial effect on *E. coli*, *P. aeruginosa* and *S. aureus*; however, this effect was less in the loaded fibers than in the free oil. This may be a consequence of interactions between the polymeric matrix fibers and oil, negatively affecting its bactericidal action.

#### 5.3.3. Memecylon Edule Extract

The use of medicinal plant extracts for the treatment of diseases or injuries is an ancient and widespread practice throughout the world. There are numerous plants described in the scientific literature that possess compounds affecting one or more of the processes related to wound healing, such as coagulation, inflammation, fibroplasia, epithelialization, collagenation, and wound contraction. Jin et al. [60] studied extracts from four types of traditional Indian medicinal plants: *Indigofera aspalathoides* (IA), *Azadirachta indica* (AI), *Memecylon edule* (ME), and *Myristica andamanica* (MA). Each of the extracts was independently mixed with the PCL polymer in a chloroform/methanol solution (50:50 *v*/*v*) in such a way as to obtain 12% solutions (*w*/*v*), and subsequently subjected to the electrospinning process to produce PCL/Extract nanofibers (PCL/IA; PCL/AI; PCL/ME; and PCL/MA). The proliferation of human dermal fibroblasts (HDF) in PCL/ME nanofibers (>80%) was significantly higher compared to the cell proliferation achieved by the PCL/IA, PCL/AI, and PCL/MA nanofibers (<35%). Biocompatibility analyses showed that PCL/ME nanofibers are much larger than the others. HDF collagen secretion was measured in the different types of nanofiber scaffolds and was found to be higher in PCL/ME than in other scaffolds. The high fibroblast proliferation rate obtained by PCL/ME fibers can be caused by triterpenes, tannins, and flavonoids present in the *Memecylon edule* plant; Jin et al. [60] reported that these compounds favor the wound healing process due to their astringent and antimicrobial properties.

The authors also carried out an assay to measure the epidermal differentiation capacity of ADSCs in PCL/ME scaffolds in which changes in fibroblast shapes could be observed, showing the existence of cell differentiation. PCL/ME-based scaffolds are a promising device for skin tissue engineering.

#### 5.3.4. Aspalathus Linearis Fermented Extract (AL Extract)

*Aspalathus linearis* is a South African plant, the leaves of which are widely used in an infusion known as rooibos tea. The plant is rich in polyphenols that have antioxidant properties. Researchers have shown that the pro-inflammatory nature of fermented rooibos may have therapeutic value for wounds characterized by a delayed initial inflammatory phase [119]. Ilomuanya et al. [20] incorporated a fermented rooibos extract, which has inherent antibacterial and healing activity, into a matrix of PLA/COL electrospun fibers for the treatment of chronic wounds and burns. The fibers loaded with AL extract showed bacterial inhibition for *S. aureus*, *P. aeruginosa*, methicillin-resistant *S. aureus,* and *E. coli*. However, this degree of inhibition is less than that shown by fibers containing AL extract/silver sulfadiazine. The fibers loaded with AL extract showed increased elimination of DPPH free radicals as the concentration in the framework increased, showing that the antioxidant activity of these fibers is due to the presence of the natural extract.

## 6. Conclusions

The different studies included in this review reported the development of electrospun fibers at nanometer scale, with high porosity, water retention capacity, and controlled release of encapsulated compounds. Several authors conclude that mechanical and physicochemical properties of both blend and core/shell fibers such as tensile strength, elasticity, flexibility, high surface-to-volume ratio, high swelling capacity, good water vapor transmission rate, and good absorption of exudate, lead to an improvement in the cellular respiration rate, the humidity regulation, and the elimination of exudates, which promote the healing process of skin burns and/or the control of infections in the affected area.

The use of natural and synthetic polymers, either individually or in combination with others, was identified. Given their similarity to physiological extracellular collagen matrix, natural polymers are demonstrated to be attractive materials for biomedical applications as they have the advantage of biological recognition and potential bioactive behavior proliferation. In fact, they are intrinsically biocompatible and biodegradable, and degradation products are well tolerated and metabolized by the human body. Regarding the synthetic materials, PCL was the most widely used (six publications), probably because of its favorable physicomechanical properties for electrospinning, its biodegradability, and its cellular biocompatibility. In addition, this review identified several types of naturally occurring bioactive compounds loaded in electrospun fibers for burn treatment. The bioactivity of these compounds could be classified into three different categories: antimicrobial; those that accelerate wound healing; and those with both biological effects. This category system is a simplified way of organizing the compounds.

Therefore, this review provided a brief overview of the development of electrospun fibers loaded with natural bioactive compounds for the treatment of skin burns with promising results in controlling infections (antimicrobial activity), biocompatibility, non-toxicity and in promoting and accelerating the natural healing process. Thanks to these advantages, this biomedical system may offer an effective alternative treatment for wound healing; in spite of this, research needs to be broadened and deepened for clinical applications in the field of the translational regenerative medicine.

## Figures and Tables

**Figure 1 pharmaceutics-13-02054-f001:**
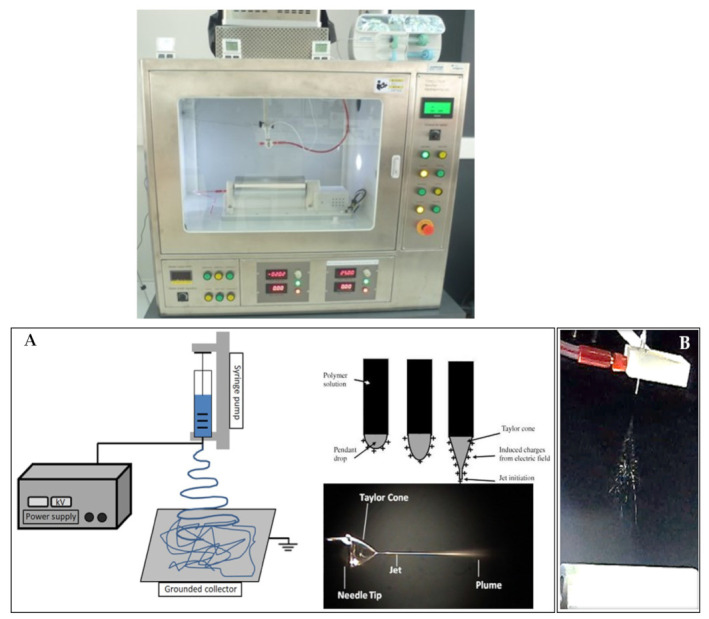
(**A**) Schematic representation of an electrospinning apparatus, showing the formation of the Taylor’s cone. Figure from Luraghi et al. [46] with permission from Elsevier. (**B**) Electrospun fiber formation.

**Figure 2 pharmaceutics-13-02054-f002:**
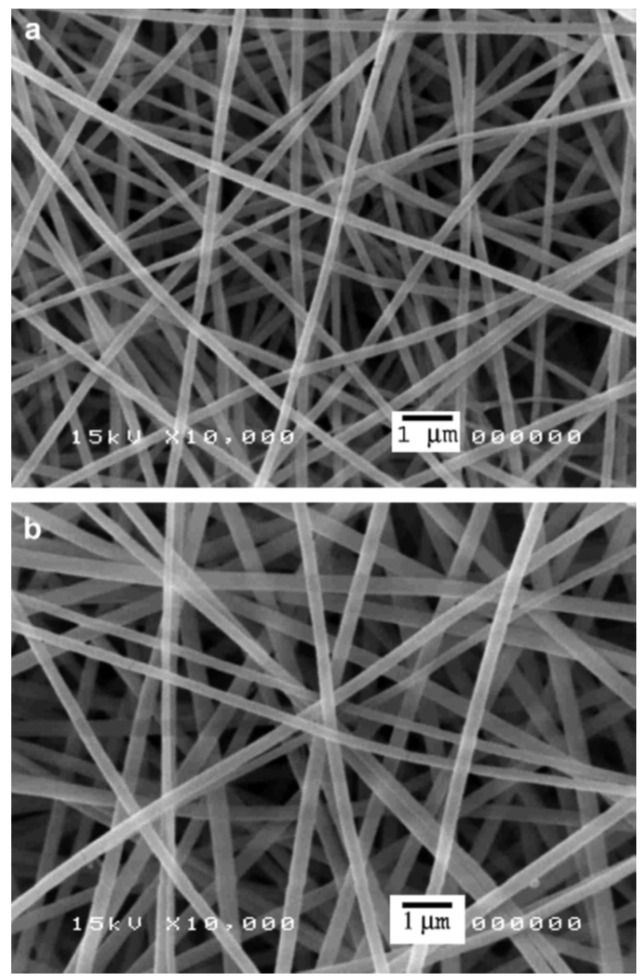
SEM images of the electrospun fiber mats from (**a**) the base gelatin solution and (**b**) the AgNO_3_-containing gelatin solution that had been aged for 12 h. Figure from Rujitanaroj et al. [27] Copyright (2021), with permission from Elsevier.

**Figure 3 pharmaceutics-13-02054-f003:**
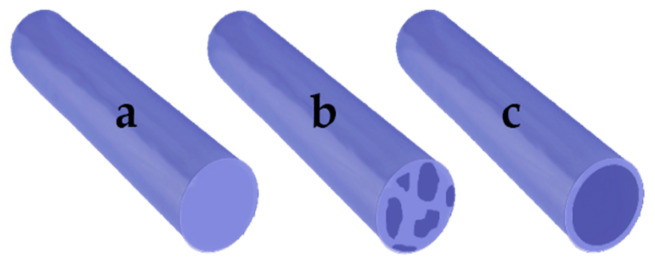
Types of electrospun fiber: (**a**) simple; (**b**) blend; (**c**) core-shell.

**Table 1 pharmaceutics-13-02054-t001:** Synthetic bioactive and/or non-biological compounds reported in the literature.

Active Compound		Reference
Metallic Particles	Silver Nitrate	[27]
	Silver NanoparticlesSilver Nitrate	[28]
	Mesoporous Silica Nanoparticles with Silver Nanoparticles	[29]
	Zinc Oxide Nanoparticles	[30,31]
	Iron oxide Nanoparticles	[31]
	Gold Nanoparticles	[31]
	Piroxicam	[32]
Synthetic compounds	Nitrofurazone	[33]
	Cefotaxime	[34]
	Hydroxyapatite	[35]
	Silver Sulfadiazine	[20,36,37]
	Peptide P12	[38]
	Polycaprolactone	[39]
	Polypropylene Fumarate	[40]
	Mineralized Magnesium	[41]

**Table 2 pharmaceutics-13-02054-t002:** Characteristics of electrospun fibers with intended biological effects from 24 articles reviewed.

Matrix	Encapsulated Bioactive Compound	Electrospinning Parameters	Diameter of Fibers	Fiber Type	Biological Effects of Electrospun Fibers	Reference
Polyurethane	Badger *(Meles meles) oil*	Voltage: 20 kV Flow Rate:—L/h Distance:15 cm	375–518 nm	Blend-composite	Antibacterial	[2]
Polyurethane/Silver nanoparticles (10/3% *w*/*w*)	Olive Oil (*Olea europaea* L.)	Voltage: 15 kV Flow Rate:—mL/h Distance: 10 cm	250–550 nm	Blend-composite	Antibacterial	[10]
Silk fibroin/Gelatin (1:3 *w*/*w*)	Astragaloside IV	Voltage: 15 kV Flow Rate: 0.1 mL/h Distance:—cm	_	Blend-composite	Accelerate the process of wound healing	[16]
Chitosan-Deacetylated	Chitosan/L-arginine	Voltage: 28 kV Flow Rate: 1.2 mL/h Distance: 10 cm	50–500 nm	Blend-composite	Antibacterial	[18]
Polycaprolactone/Gelatin (Core) (8/4% *w*/*w*) Gelatin (Shell)	Minocycline hydrochloride *G. sylvestre* extracts	Voltage: 13 kV Flow Rate: 1.2 & 1 mL/h Distance: 12 cm	300–450 nm	Core/Shell	Antibacterial Nanofibers	[19]
Polylactide/Collagen (20/4% *w*/*v*)	Fermented rooibos *A. linearis* extracts	Voltage: 25 kV Flow Rate: 0.1 mL/min Distance: 22 cm	13–23 µm	Blend-composite	Antibacterial Nanofibers; Accelerate the process of wound healing	[20]
Chitosan	Bromelain	Voltage: 10 kV Flow Rate: 0.5 mL/h Distance: 20 cm.	140–360 nm	Blend-composite	Accelerate the process of wound healing	[21]
Polylactide/Poly(ethylene glycol) (Core) (1:1 *w*/*w*) Polylactide/Poly(vinyl pyrrolidone) (Shell) (5:5, 7:3, 8:2, 9:1 *w*/*w*)	Peptides HHC36 Curcumin	Voltage: 20 kV Flow Rate:—mL/h Distance: 15 cm	3.2–4.6 μm	Core/Shell	Antibacterial	[22]
Gelatin	ε-Polylysine	Voltage: 12 kV Flow Rate: 0.8 mL/h Distance: 12 cm	425 ± 33 nm	Blend-composite	Antibacterial	[24]
Poly(vinyl alco-hol)	Chitosan	Voltage: 18 kV Flow Rate: 0.8 mL/h Distance: 12 cm	130–170 nm	Blend-composite	Antibacterial	[25]
Poly(3-hydroxybutyrate-co-3-hydroxyvalerate)	_	Voltage: 8 kV Flow Rate: 0.002 mL/min Distance: 12 cm	510–670 nm	Simple Fibers (Mono-polymer)	Accelerate the process of wound healing	[26]
Silk fibroin/Poloxamer 407 (P407) (1:0, 3:1, 1:1 *w*/*w*)	Manuka Honey	Voltage: 25–23 kV Flow Rate: 3–4 mL/h Distance: 16.5–18 cm	2.4–5.9 μm	Blend-composite	Antibacterial; Accelerate the process of wound healing	[47]
Sodium Alginate-Poly(ethylene glycol)/Pluronic F127 (surfactant) (8:2 *w*/*w*—1.5% *w*/*v*)	Lavender essential oil *(Lavandula angustifolia)*	Voltage: 25 kV Flow Rate: 0.5 mL/h Distance: 20 cm	50–125 nm	Blend-Emulsion Electrospinning	Antibacterial Nanofibers; Accelerate the process of wound healing	[48]
Polycaprolactone/Chitosan (10, 15, 20/15% *w*/*w*)	Quercetin/Rutin	Voltage: 24–32 kV Flow Rate: 0.77 mL/h Distance: 15 cm	90–120 nm	Blend-composite	Antibacterial Nanofibers; Accelerate the process of wound healing	[55]
Chitosan/Poly(ethylene oxide) (2/0.5% *w*/*w*)	Actinidin	Voltage:—kV Flow Rate: 0.5–1.5 mL/h Distance: 7–9 cm	100–200 nm	Blend-composite + Actinidin enzyme immobilization	Antibacterial Nanofibers; Accelerate the process of wound healing	[56]
Gelatin (layer 1) Poly(vinyl alcohol)/Sodium Alginate (layer 2) (13/2.5% *w*/*v*) Chitosan/Poly(vinyl alcohol) (layer 3) (2/15% *w*/*v*)	Fibrin	Voltage: 25–30 kV Flow Rate: 0.8–1.1 mL/h Distance:—cm	150–350 nm	Blend-composite	Antibacterial Nanofibers; Accelerate the process of wound healing	[57]
Polycaprolactone	α-Lactalbumin	Voltage: 9–18 kV Flow Rate: 0.3–0.6 mL/min Distance: 15 cm	183–344 nm	Blend-composite	Accelerate the process of wound healing	[58]
Poly(vinyl pyrrolidone)/Keratin (3:1, 2:1, 1:1 *w*/*w*)	Cinnamon essential oil	Voltage: 24 kV Flow Rate: 350–850 μL/h Distance: 25 cm	315–466 nm	Blend-composite	Antibacterial Nanofibers; Accelerate the process of wound healing	[59]
Polycaprolactone/Gelatin (6:4 *w*/*w*)	Plant extracts: *I. aspalathoides A. indica* *M. edule* *M. andamanica*	Voltage: 15 kV Flow Rate: 1 mL/h Distance: 12 cm	266–601 nm	Blend-composite	Accelerate the process of wound healing	[60]
Poly-D,L-lactic acid	Microalga Spirulina (*Arthrospira* *platensis*)	Voltage: 15 kV Flow Rate: 2 mL/h Distance: 15 cm	260–270 nm	Blend-composite	Accelerate the process of wound healing	[61]
Poly(L-lactic acid)/polyhedral oligomeric silsesquioxane nanoparticles (24:1 *w*/*w*)	Plasmid DNA Encoding Angiopoietin-1 (pAng)	Voltage: 13 kV Flow Rate: 0.8 mL/h Distance: 15 cm	580–780 nm	Blend-composite	Accelerate the process of wound healing	[62]
Polycaprolactone/Collagen (55:25 *w*/*v*)	_	Voltage: 13 kV Flow Rate: 3 mL/h Distance: 13 cm	170–275 nm	Blend-composite	Accelerate the process of wound healing	[63]
Poly(lactic-co-glycolic acids)/Collagen (4:1 *w*/*w*)	_	Voltage: 28 kV Flow Rate: 1 mL/h Distance: 17 cm	100–300 nm	Blend-composite	Accelerate the process of wound healing	[64]
Polycaprolactone (12.5% *w*/*v*) Poly(vinyl al-co-hol) (8% *w*/*v*)	Curcumin	Voltage: 12, 18, 24 kV Flow Rate: 1, 2, 3 mL/h Distance: 16 cm	_	Blend-composite	Antibacterial	[65]

**Table 3 pharmaceutics-13-02054-t003:** Materials used for the production of electrospun fibers in 24 articles reviewed.

Material Type	Material Name	Reference
Natural	Chitosan	[18,21,55,56,57]
	Collagen	[20,63,64]
	Gelatin	[19,57,60,66,67]
	Keratin	[59]
	Poly(3-hydroxybutyrate-co-3-hydroxyvalerate)	[26]
	Silk Fibroin	[16]
	Sodium Alginate	[48,57]
Synthetic	Poly(ethylene glycol)/Poly(ethylene oxide)/Polyoxyethylene	[22,56]
	Poly(lactic-co-glycolic acids)	[64]
	Poly(L-lactic acid)	[62]
	Poly(vinyl alcohol)	[25,57,65]
	Poly(vinyl pyrrolidone)	[22,59]
	Polycaprolactone	[19,55,58,60,63,65]
	Poly-D,L-lactic acid	[61]
	Polylactide	[20,22]
	Polyurethane	[2]

**Table 4 pharmaceutics-13-02054-t004:** Natural bioactive compounds with antimicrobial properties and/or which accelerate burn wound healing loaded in electrospun fibers.

Biological Effects	Bioactive Compounds	Reference
Antimicrobial	Badger *(Meles meles)* oil	[2]
	Olive (*Olea europaea* L.) oil	[16]
	CH—CH/L-arginine	[18,25]
	*Gymnema sylvestre* extract	[19]
	ε-Polylysine	[24]
	Manuka Honey (*)	[47]
	Peptides HHC36	[62]
	Curcumin	[62,65]
Wound healing accelerator	Astragaloside IV	[16]
	Bromelain	[21]
	Quercetin/Rutin	[55]
	Actinidin	[56]
	Fibrin	[57]
	α-Lactalbumin	[58]
	Microalga Spirulina (*Arthrospira platensis*)	[61]
	Plasmid DNA Encoding Angiopoietin-1	[62]
Antimicrobial and wound healing accelerator	*Aspalathus linearis* fermented extract	[20]
	Lavender *(Lavandula angustifolia)* essential oil	[48]
	Cinnamon (*Cinnamomum verum*) essential oil	[59]
	*Indigofera aspalathoides* extract	[60]
	*Azadirachta indica* extract	[60]
	*Memecylon edule* extract	[60]
	*Myristica andamanica* extract	[60]

(*) Used in the study for its hygroscopic capacities, not for its antimicrobial activity.

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
