# Peer review of "Electrospun Fibers Loaded with Natural Bioactive Compounds as a Biomedical System for Skin Burn Treatment. A Review"

_pharmaceutics, 2021, doi:10.3390/pharmaceutics13122054_

Round 1
Reviewer 1 Report
1. In the abstract, it is stated, "In conclusion, electrospun membranes loadedwith natural bioactive compounds could be an efficient and economically viable
alternative for skin burn treatment; however, research needs to be broadened and
deepened to obtain stronger results." However, the conclusions section
seems to be part of a discussion and did not support the stated on the abstract.
2. References must be updated, I suggest to revised the review article
(https://doi.org/10.1155/2021/9920755).
Author Response
Point 1. In the abstract, it is stated, "In conclusion, electrospun membranes loaded with natural bioactive compounds could be an efficient and economically viable alternative for skin burn treatment; however, research needs to be broadened and deepened to obtain stronger results." However, the conclusions section seems to be part of a discussion and did not support the stated on the abstract.
Reponse 1. The reviewer´s comment is right. We deleted the "discussion part" of the conclusion section. The abstract was also restructured.
Point 2. References must be updated; I suggest to revised the review article
(https://doi.org/10.1155/2021/9920755).
Reponse 2. The references were updated. We included the following references:
- Villarreal-Gómez, L. J., Pérez-González, G. L., Bogdanchikova, N., Pestryakov, A., Nimaev, V., Soloveva, A., Toledaño-Magaña, Y., et al. Antimicrobial Effect of Electrospun Nanofibers Loaded with Silver Nanoparticles: Influence of Ag Incorporation Method. Journal of Nanomaterials, 2021.
Reviewer 2 Report
General comments:
The influence of electrospinning parameters on fibers morphology and the methodology for core-shell fibers should be explained. Moreover, as the review describes electrospinning the scheme of such a device is applicable. The authors should include SEM images of electrospun fibers to show how the morphology looks like. This is important, as the authors mentioned fibers similarity to the ECM. In the text, there is a lot of materials characterizations, but there is no information about materials requirements (should be mentioned in the introduction). Polymer’s characterization is not consistent, some of them consists of the type of polymerization or molecular weight, some not, please standardize it. In the section 5.1 for some of bioactive compounds only one example of electrospun fibers is mentioned, this is insufficient. Furthermore, this section should be divided for the type of bioactive compound, for example: proteins, oils, plants, plants extracts, etc.. There are a lot of misspellings of PCL, authors wrote PLC. Conclusion section should be more about advantages and disadvantages about different wound healing system. More, the comparison of presented studies should be done. It is unnecessary to mention for example all bioactive substances again. Please correct references number: 3, 6, 11, 19, 26, 29, 33, 34, 35, 36, 37, 61, 64, 68, 69, 79, 88, 94 and 102. Summarizing, this review needs a major correction and in such a form should be rejected, in particular the part of the electrospinning. More, sometimes the language is poor and sentences are difficult to understand.
Page 1 line 19, the word “artefacts” is misleading.
Page 2 line 32, the sentence “series of varied” redundant, series or varied would be enough.
Page 2 line 33, the word “stage” is wrong. Skin healing process has many stages.
Page 2 line 51, electrospun fibers diameters range from nanometers to micrometers, so the fibers have a wider range of diameter.
Page 2 line 53, could you explain how the electrospun fibers itself prevents infections.
Page 2 line 59, does the polymer concentration really affects the fibers diameter, what’s more?
Page 2 line 85, table 1. Is it necessary to create a table and cite synthetic compounds, as the review refers to the natural ones?
Page 3 line 86, rename the heading of section 3. The order on the sentence is not correct.
Page 3 line 91, does the collector always have opposite electrical charge, when compared to the needle? In many cases the collector is grounded.
Page 3 line 107, repetition, “Simple fibers: these are the simplest…”
Page 3 line 121, “Core/shell fibers: core-shell electrospinning (also called co-axial electrospinning) involves electrospinning a drug/polymer blend as a core material entrapped within a shell produced from a second polymer solution.” Is it possible to create a system with drug in the shell?
Page 4 line 135, “In general, the articles reviewed…generation of injured skin tissue [17].” This is a crucial information in this review, thus it should go in the beginning of this section.
Page 4 line 141, when the authors refer to blends, the ratio should be mentioned.
Page 8, line 226, authors wrote “PLA since it is a biocompatible…” but in the further authors said “To improve its biocompatibility…” Is the PLA biocompatible or not, please clarify it.
Page 9 line 295, the word “produced” is unnecessary here.
Page 9 line 319, FDA approves devices, drug etc. rather than material itself.
Page 9 line 321, Please clarify this sentence, “and its suitability for export to countries and cultures where the implantation of animal products is unpopular [44].”
Page 9 line 322, “Sadeghi-Avalshahr et al. [28] used PLGA to” it has been already explained in the section about collagen (4.1.2)
Page 10 line 354, there is a double space between the words “lower” and “synthetic”.
Page 11 line 384, “Finally, Saeed et al. [49] developed a multilayer device of PCL and PVA.” It has already been explained in the section about PVA (4.2.4).
Page 11 line 418, authors mentioned the stages of wound healing, this information is crucial for this review, thus it should be mentioned in the beginning of this work.
Page 12 line 431, what was the purpose to use badger oil in the cited study? The authors claim that the antibacterial effect come from Ag, rather than oil.
Page 12 line 447, is chitosan a bioactive compound in this context? The 5.1.2 section should be combined with 4.1.1 section.
Page 15 line 590, authors claim: “that highly porous scaffolds are produced when it is mixed with other polymers,”, it suggest that porous structure is obtained, when polymers are mixed, which is not a true.
Page 17 line 656, the word “haemostasis” is misspelled.
Page 17 line 699, the heading 5.3. should be corrected, for example: Natural bioactive compounds with antimicrobial effects for burn wound healing.
Page 18 line 712, “The fibers efficiently released the nanofibers, peaking after 6 hours.” Please correct this sentence.
Page 18 line 713, “The SA-PEO/lavender oil fibers presented hybridization of bacterial colonies of S. aereus” please clarify this sentence. What hybridization means in this context?
Page 18 line 716, “achieving a decrease in IL-6 (66%) and IL-8 (49%) concentrations”, please clarify this, was the IL-6 and IL-8 concentration measured during in vitro or in vivo study. This sentence is incomplete.
Page 18 line 738, “These good results of PLC” too broadly explained.
Page 19 line 778, “The various authors conclude that the mechanical characteristics of both blend and core/shell fibers effectievly…”, what does it mean? What are the preferable mechanical properties of electrospun fibers for wound healing patches? How do they affect the wound healing process and control infections?
Author Response
Reviewer 2
Point 1. The influence of electrospinning parameters on fibers morphology and the methodology for core-shell fibers should be explained. Moreover, as the review describes electrospinning the scheme of such a device is applicable. The authors should include SEM images of electrospun fibers to show how the morphology looks like. This is important, as the authors mentioned fibers similarity to the ECM. In the text, there is a lot of materials characterizations, but there is no information about materials requirements (should be mentioned in the introduction). Polymer’s characterization is not consistent, some of them consists of the type of polymerization or molecular weight, some not, please standardize it. In the section 5.1 for some of bioactive compounds only one example of electrospun fibers is mentioned, this is insufficient. Furthermore, this section should be divided for the type of bioactive compound, for example: proteins, oils, plants, plants extracts, etc. There are a lot of misspellings of PCL, authors wrote PLC. Conclusion section should be more about advantages and disadvantages about different wound healing system. More, the comparison of presented studies should be done. It is unnecessary to mention for example all bioactive substances again. Please correct references number: 3, 6, 11, 19, 26, 29, 33, 34, 35, 36, 37, 61, 64, 68, 69, 79, 88, 94 and 102. Summarizing, this review needs a major correction and in such a form should be rejected, in particular the part of the electrospinning. More, sometimes the language is poor and sentences are difficult to understand.
Response 1: Thank you very much for your comments.
An SEM image of electrospun fibers (Figure 3) was added as an example.
The information about materials requirements was included.
Regarding your comments, we standardized the section 4.
About your suggestion to rearrange the classifications of bioactive compounds, we believe that the following classification is correct: antimicrobial bioactive, accelerating wound healing, and those with both capabilities.
We correct the term PCL through the whole text.
Regarding the Conclusion section, we deleted the paragraph of bioactive substances. We simplified the conclusion for an easier comprehension.
We corrected the references numbers 3, 6, 11, 19, 26, 29, 33, 34, 35, 36, 37, 61, 64, 68, 69, 79, 88, 94 and 102.
Point 2. Page 1 line 19, the word “artefacts” is misleading.
Response 2: We deleted the word “artefacts” through the text.
Point 3. The sentence “series of varied” redundant, series or varied would be enough.
Response 3: In fact the sentence “series of varied” is redundant. This sentence was changed in the whole text.
Point 4. Page 2 line 33, the word “stage” is wrong. Skin healing process has many stages.
Response 4: Thanks for the comment. We changed the sentence by “Burned skin undergoes varied and complex processes after trauma, including coagulation, inflammation, cell proliferation, and tissue remodeling.”
Point 5. Electrospun fibers diameters range from nanometers to micrometers, so the fibers have a wider range of diameter.
Response 5. We greatly appreciate your comment We changed the sentence by “The diameters of these fibers range from a few nanometers to micrometers”.
Point 6. Page 2 line 53, could you explain how the electrospun fibers itself prevents infections.
Response 6. Based on the literature, electrospun fibers could prevent infections in two different ways:
- To act as a physical barrier that prevents or minimizes the migration of microorganisms to the affected area, as happens with wound dressings [1-3]
- The second and most interesting way is that the material used for the fiber production possesses antimicrobial activity, such as chitosan (a polymer widely used in electrospinning with high hydrophobicity, a characteristic to which its antibacterial activity is attributed) [4-5].
- Rath, G., Hussain, T., Chauhan, G., Garg, T., and Kumar Goyal, A. Fabrication and characterization of cefazolin-loaded nanofibrous mats for the recovery of post-surgical wound. Artificial cells, nanomedicine, and biotechnology 2016, 44(8), 1783-1792.
- Liu, M., Duan, X. P., Li, Y. M., Yang, D. P., and Long, Y. Z. Electrospun nanofibers for wound healing. Materials Science and Engineering: C 2017, 76, 1413-1423.
- Zou, P., Lee, W. H., Gao, Z., Qin, D., Wang, Y., Liu, J., Gao, Y., et al. Wound dressing from polyvinyl alcohol/chitosan electrospun fiber membrane loaded with OH-CATH30 nanoparticles. Carbohydrate polymers 2020, 232, 115786.
- Ignatova, M., Manolova, N., and Rashkov, I. Electrospun Antibacterial Chitosan‐B ased Fibers. Macromolecular bioscience 2013, 13(7), 860-872.
- Adeli, H., Khorasani, M. T., and Parvazinia, M. Wound dressing based on electrospun PVA/chitosan/starch nanofibrous mats: Fabrication, antibacterial and cytocompatibility evaluation and in vitro healing assay. International journal of biological macromolecules 2019, 122, 238-254.
Point 7. Page 2 line 59, does the polymer concentration really affects the fibers diameter, what’s more?
Response 7: Thank you for your comment. Indeed this paragraph lacks information about the factors that affect the fiber diameter. Therefore the paragraph “The concentration of polymer in a solution determines whether it can be electrospun into nanofibers, and moreover has an important effect on fiber morphology” was replaced by “Factors such as the polymer concentration, viscosity, elasticity, polarity and conductivity of the solution determine whether it can be electrospun into fibers, and also have an important effect on their morphology [8,103]". This modification was done in page 2 lines 66-69. Additionally, the following reference was added:
- Keirouz, A., Chung, M., Kwon, J., Fortunato, G., and Radacsi, N. 2D and 3D electrospinning technologies for the fabrication of nanofibrous scaffolds for skin tissue engineering: A review. Wiley Interdisciplinary Reviews: Nanomedicine Nanobiotechnology 2020, 12(4), e1626.
Point 8. Page 2 line 85, table 1. Is it necessary to create a table and cite synthetic compounds, as the review refers to the natural ones?
Response 8: We understand your comment, however we believe that it is necessary to contextualize in the text that different compounds can be encapsulated in electrospun fibers for therapeutic burn purposes: natural compounds (which are discussed in this paper) and synthetic compounds that are summarized in the Table 1.
Point 9. Page 3 line 86, rename the heading of section 3. The order on the sentence is not correct.
Response 9: The heading of section 3 “Electrospinning: Categorization and characterization of the types of fibers obtained and used” was replaced by “Categorization and characterization of electrospun fibers”
Point 10. Page 3 line 91, does the collector always have opposite electrical charge, when compared to the needle? In many cases the collector is grounded.
Response 10: We greatly appreciate your comment. Indeed the collector does not always have an opposite charge, in fact it may sometimes be connected to ground. The sentence "the polymeric solution travels to a collector with opposite electrical charge [13]." was replaced by “the polymer solution travels to a collector that may have an opposite electrical charge or be grounded [13, 104] (Figure 1). Additionally, the following reference was added:
- Zheng, G., Jiang, J., Wang, X., Li, W., Yu, Z., and Lin, L. High-aspect-ratio three-dimensional electrospinning via a tip guiding electrode. Materials & Design 2021, 198, 109304.
Point 11. Page 3 line 107, repetition, “Simple fibers: these are the simplest…”
Response 11: As you mention, the phrase "Simple fibers: these are the simplest and most basic type of electrospun fiber." is repetitive, so it was replaced by " Simple fibers: these are the most basic type of electrospun fiber. "
Point 12. Page 3 line 121, “Core/shell fibers: core-shell electrospinning (also called co-axial electrospinning) involves electrospinning a drug/polymer blend as a core material entrapped within a shell produced from a second polymer solution.” Is it possible to create a system with drug in the shell?
Response 12: Yes, in the core / shell electrospinning method, both core and shell can be loaded with drug. For further clarification, the sentence “Core / shell fibers: core-shell electrospinning (also called coaxial electrospinning) involves electrospinning a drug / polymer blend as a core material entrapped within a shell produced from a second polymer solution" was replaced by “Core/shell fibers: Core/shell electrospinning, also called coaxial electrospinning, is a modification of conventional electrospinning, characterized by the use of sample ejection capillaries arranged for the injection of a solution into the other solution. The core/shell fibers have two clearly different sections, a central core formed by a solution, and a shell or outer layer formed by another solution [105]. Core and shell can encapsulate drugs independently [6, 16, 19].”
Additionally, the following reference was added:
- Qin, X. "Coaxial electrospinning of nanofibers." Electrospun nanofibers. Woodhead Publishing, 2017. 41-71.
Point 13. Page 4 line 135, “In general, the articles reviewed…generation of injured skin tissue [17].” This is a crucial information in this review, thus it should go in the beginning of this section.
Response 13: We appreciate your comment and suggestion, however we consider that such information could have a greater impact by moving the paragraph in the introduction, rather than at the beginning of section 3.
Point 14. Page 4 line 141, when the authors refer to blends, the ratio should be mentioned.
Response 14. The ratios are mentioned in Table 2.
Point 15. Page 8, line 226, authors wrote “PLA since it is a biocompatible…” but in the further authors said “To improve its biocompatibility…” Is the PLA biocompatible or not, please clarify it.
Response 15: Thanks for mentioning this point. It is important to clarify that polylactic acid (PLA) is a biocompatible polymer widely used for biomedical applications. The contradiction in the text is due to a typing error. Therefore, the phrase “To improve its biocompatibility and mechanical properties it must be mixed with another polymer - in this case the authors used COL” was replaced by “To improve its mechanical properties, it must be mixed with another polymer, in this case the authors used collagen.”
Point 16. Page 9 line 295, the word “produced” is unnecessary here.
Response 16. We agree that the word is unnecessary so it was deleted from the text.
Point 17. Page 9 line 319, FDA approves devices, drug etc. rather than material itself.
Response 17: We agree that the phrase “FDA approval for clinical use in humans” is inaccurate. As you mention the FDA does not approve materials themselves. Consequently, it was replaced by the phrase “the approval by the Food and Drug Administration (FDA) of more than 20 PLGA-based pharmaceutical products to date [106];
Additionally, the following reference was added:
- Wang, Y., Qin, B., Xia, G., and Choi, S. H. FDA’s poly (lactic-co-glycolic acid) research program and regulatory outcomes. The AAPS Journal 2021, 23(4), 1-7.
Point 18. Page 9 line 321, Please clarify this sentence, “and its suitability for export to countries and cultures where the implantation of animal products is unpopular [44].”
Response 18: Analyzing the phrase "and its suitability for export to countries and cultures where the implantation of animal products is unpopular [44].", We agree that it is confusing and does not provide relevant information, so it was deleted.
Point 19. Page 9 line 322, “Sadeghi-Avalshahr et al. [28] used PLGA to” it has been already explained in the section about collagen (4.1.2).
Response 19: The sentence was modified to not repeat the information. The new paragraph is “Sadeghi-Avalshahr et al. [28] used PLGA to produce electrospun nanofibers, which are characterized by high hydrophobicity. This could negatively affect the efficiency of cell growth and adhesion. The authors mixed PLGA with collagen using two methods, as de-scribed in section 4.1.2, since this natural polymer would help make the fibers less hydrophobic.
Point 20. Page 10 line 354, there is a double space between the words “lower” and “synthetic”.
Response 20: The error was corrected in the text.
Point 21. Page 11 line 384, “Finally, Saeed et al. [49] developed a multilayer device of PCL and PVA.” It has already been explained in the section about PVA (4.2.4).
Response 21: We agree that the phrase “Finally, Saeed et al. [49] developed a multilayer device of PCL and PVA. The external layers were made of PCL loaded with curcumin, and in addition to the mechanical properties provided by the polymer, controlled release of curcumin was sought ” was already explained in the section (4.2.4), so it was replaced by“ The role of PCL in the PCL / PVA / PCL multilayer devices reported by Saeed et al. [49] was to load the curcumin and to provide the mechanical properties for a controlled release of curcumin ”.
Point 22. Page 11 line 418, authors mentioned the stages of wound healing, this information is crucial for this review, thus it should be mentioned in the beginning of this work.
Response 22. We agree with your comment, so we have modified the sentence in the text and place it to section 1 of the introduction. The sentence " The wound healing process can be achieved through the individual or combined action of bioactive agents and/or agents that promote angiogenesis, epithelialization, collagenation or wound contraction “ is located now on page 2, lines 95-97.
Point 22. Page 12 line 431, what was the purpose to use badger oil in the cited study? The authors claim that the antibacterial effect come from Ag, rather than oil.
Response 22: The authors specify that badger oil improved the crosslinking of the PU fiber network, so its role was to improve the mechanical properties of the fibers. Despite the fact that badger oil has antibacterial action according to the literature, this could not be observed in the trials of this research.
Point 23. Page 12 line 447, is chitosan a bioactive compound in this context? The 5.1.2 section should be combined with 4.1.1 section.
Response 23: Chitosan is a material that has the ability to be used both as polymer and/or bioactive. In this work, both functions are differentiated by separating them into different sections to avoid confusion for readers. In the section 5.1.2 we specified the use of chitosan as a bioactive, and in the section 4.1.1 we detailed the use of chitosan as a polymeric material for the development of electrospun fibers. Therefore, we prefer not to combine the sections 4.1.1 with 5.1.2.
Point 24. Page 15 line 590, authors claim: “that highly porous scaffolds are produced when it is mixed with other polymers,”, it suggest that porous structure is obtained, when polymers are mixed, which is not a true.
Response 24. As you mentioned, the phrase "that highly porous scaffolds are produced when it is mixed with other polymers ..." is not correct, so it was removed from the text.
Point 25. Page 17 line 656, the word “haemostasis” is misspelled.
Response 25: The word “haemostasis” was corrected to “hemostasis”.
Point 26. Page 17 line 699, the heading 5.3. should be corrected, for example: Natural bioactive compounds with antimicrobial effects for burn wound healing.
Response 26: Thanks for the comment. We have changed the heading 5.3 to “5.3. Natural bioactive compounds with antimicrobial effects and which accelerate burn wound healing” fue reemplazado por “5.3. Natural bioactive compounds with antimicrobial effects for burn wound healing”.
Point 27. Page 18 line 712, “The fibers efficiently released the nanofibers, peaking after 6 hours.” Please correct this sentence.
Response 27: The sentence was corrected to "The fibers efficiently released the oil, peaking after 6 hours."
Point 28. Page 18 line 713, “The SA-PEO/lavender oil fibers presented hybridization of bacterial colonies of S. aereus” please clarify this sentence. What hybridization means in this context?
Response 28. The word "hybridization" is misspelled. The correct word is "inhibition". The error was corrected in the tex.
Point 29. Page 18 line 716, “achieving a decrease in IL-6 (66%) and IL-8 (49%) concentrations”, please clarify this, was the IL-6 and IL-8 concentration measured during in vitro or in vivo study. This sentence is incomplete.
Response 29: This analysis was performed under in vitro conditions. This information was added in the sentence.
Point 30. Page 18 line 738, “These good results of PLC” too broadly explained.
Response 30: We agree that the sentence “These good results of PCL at the cellular level may be…” is too broad, so it was replaced by “The high fibroblast proliferation rate obtained by PCL / ME fibers can be…”.
Point 31. Page 19 line 778, “The various authors conclude that the mechanical characteristics of both blend and core/shell fibers effectievly…”, what does it mean? What are the preferable mechanical properties of electrospun fibers for wound healing patches? How do they affect the wound healing process and control infections?
Response 31: We complete the information as follows: Several authors conclude that mechanical and physicochemical properties of both blend and core/shell fibers such as tensile strength, elasticity, flexibility , high surface-to-volume ratio, high swelling capacity, good water vapor transmission rate, good absorption of exudate, lead to an improvement in the rate of cellular respiration, the regulation of humidity and the elimination of exudates, which promote the healing process of skin burns and/or the control of infections in the affected area.

Reviewer 3 Report
Comments to the authors.
The manuscript entitled "Electrospun fibers loaded with natural bioactive compounds as a biomedical system for skin burn treatment. A review." Jeyson Hermosilla et al. have shown in their manuscript the fibers obtained from electrospinning in different systems and/or conditions, i.e., different combinations of macromolecules (polymer and proteins), additives that promote some specifically action fibers such as antimicrobial and accelerate the process of wound healing, as well different types of fibers such as simple fibers (just one macromolecule), blend/composite fibers (association of two or more macromolecules/molecule/material), and core-shell fibers (fibers of some macromolecules capped with other macromolecules/molecules). The manuscript took into account forty-nine publications for the years 2006 to 2021. These manuscripts were analyzed and categorized the physicochemical and structural properties of fibers with therapeutic potential for treating skin burns. The English grammar must be revised. The manuscript is able for publication in the Pharmaceutics journal, after minor revision. After a critical evaluation of the manuscript, some comments are made as follows:
1- The keyword must be different from the title. It enhances the research field.
2-Why in lines 71-73 does the author mentioning only one manuscript? What does this manuscript have in particular comparing with other manuscripts? Please, include more information about it.
3- Please, standardize the names in Table 1.
4- According to Table 1, more than 16 publications were utilized. Please check it.
5- The phrase in lines 88-89, "The electrospinning method consists of pumping a simple or complex polymer solution through a capillary subjected to a high-voltage electric field." must be referenced.
6- Why did the author not mention melting electrospinning?
7- Line 101: the reference Li et al. [17] did not include in Table 2, as mentioned in the manuscript. Also, in Table 2, we can observe three references that present diameters in micrometers, being references number 15, 16, and 29. There are two reference numbers 18 in Table 2. Please check it.
8- What meaning of "O/W" in line 120? Please, include this information in the manuscript.
9- Please include the complete name before the acronyms/abbreviations/label, after include it uses/applies the acronyms/abbreviations/label. It must be made/applied in the manuscript.
10- According to Table 2 is not presented 15 articles reviewed. Please check it.
11- Please, include a description of the acronyms/abbreviations/label used in Table 2.
12- The authors must evaluate the difference between blend and composite critically.
13- Collagen, gelatin, keratin, and silk fibroin are proteins and not polymers. Please, revise it in the manuscript.
14- Although many studies consider chitosan a natural polymer, chitosan is a semi-synthetic polymer because it is from the deacetylation of chitin.
15- Please, include the correct form of the chitosan units. I recommend a good reference: 10.3390/pharmaceutics13050621.
16- What meaning of: HFIP (line 199), DPPH (line 239), FDA (line 319), TFA (line 459), DCM (line 460), SEM (line482), FTIR (line 572), DSC (line 573), Ch (line 575), EDC/NHS (line 669), PBS (line 669- please, include the pH), PLC (line 730).
17- The phrase in lines 573-574: "DSC also confirmed that bromelain is encapsulated in nanofibers" is necessary?
18- The English grammar must be revised.
19- There is a lack of some Figures/images to comprehend better electrospinning apparatus. Also, the authors could include a Figure/image considering the fibers' applications in treatments of skins burns.
Author Response
Reviewer 3
The manuscript entitled "Electrospun fibers loaded with natural bioactive compounds as a biomedical system for skin burn treatment. A review." Jeyson Hermosilla et al. have shown in their manuscript the fibers obtained from electrospinning in different systems and/or conditions, i.e., different combinations of macromolecules (polymer and proteins), additives that promote some specifically action fibers such as antimicrobial and accelerate the process of wound healing, as well different types of fibers such as simple fibers (just one macromolecule), blend/composite fibers (association of two or more macromolecules/molecule/material), and core-shell fibers (fibers of some macromolecules capped with other macromolecules/molecules). The manuscript took into account forty-nine publications for the years 2006 to 2021. These manuscripts were analyzed and categorized the physicochemical and structural properties of fibers with therapeutic potential for treating skin burns. The English grammar must be revised. The manuscript is able for publication in the Pharmaceutics journal, after minor revision. After a critical evaluation of the manuscript, some comments are made as follows:
Point 1- The keyword must be different from the title. It enhances the research field.
Response 1. We appreciate your comment. Now, the new keywords are: antimicrobial agents, electrospinning, wound healing, biomaterials.
Point 2 -Why in lines 71-73 does the author mentioning only one manuscript? What does this manuscript have in particular comparing with other manuscripts? Please, include more information about it.
Response 2: We have modified the sentence and added more references “The micro and nanofiber devices developed by electrospinning as economically viable and physicochemically promising in the treatment of burns [12,19, 21, 29].”
Point 3- Please, standardize the names in Table 1.
Response 3. The names in Table 1 were standardized.
Point 4. According to Table 1, more than 16 publications were utilized. Please check it.
Response 4: we have checked the Table 1 and now there are exactly 16 publications (29, 88, 89, 90, 91, 92, 93, 94, 95, 96, 97, 98, 99, 100, 101 y 102).
Point 5. The phrase in lines 88-89, "The electrospinning method consists of pumping a simple or complex polymer solution through a capillary subjected to a high-voltage electric field." must be referenced.
Response 5. We added more references [6, 89, 94].
Point 6. Why did the author not mention melting electrospinning?
Response 6. We greatly appreciate your comment. We have not included melting electrospinning in this review since it was not used by the investigations reviewed for this review.
Point 7. Line 101: the reference Li et al. [17] did not include in Table 2, as mentioned in the manuscript. Also, in Table 2, we can observe three references that present diameters in micrometers, being references number 15, 16, and 29. There are two reference numbers 18 in Table 2. Please check it.
Response 17: The reviewer´s comment is right. We replaced “… the studies of Kadakia et al. [16] and Li et al. [17] (Table 2)” by “…the studies of Kadakia et al. [15], Li et al. [16] and Ilomuanya et al. [29] (Table 2).”
In fact, the Table 2 showed two references 18. We have replaced the number 18 by number 17 (the correct number of the reference).
Point 8. What meaning of "O/W" in line 120? Please, include this information in the manuscript.
Response 8: O/W was defined in the text. This acronym means Oil/Water.
Point 9. Please include the complete name before the acronyms/abbreviations/label, after include it uses/applies the acronyms/abbreviations/label. It must be made/applied in the manuscript.
Response 9. We appreciate your comment. In fact we have modified the acronyms and abbreviations throughout the text, to avoid confusion for readers.
Point 10. According to Table 2 is not presented 15 articles reviewed. Please check it.
Response 10: The reviewer´s comment is correct. The Table 2 summarizes the 24 publications analyzed, and not 15 as mentioned in the title of Table 2. That error was corrected.
Point 11. Please, include a description of the acronyms/abbreviations/label used in Table 2.
Response 11: All acronyms in Table 2 were replaced by full names as recommended.
Point 12- The authors must evaluate the difference between blend and composite critically.
Response 12. Indeed we have not comment about the difference between Blend-fiber and Composite-fibers. This information was incorporated in page 4 lines 145-152. Additionally, the following references were added to the review:
110 Buzgo, M., Mickova, A., Rampichova, M., and Doupnik, M. Blend electrospinning, coaxial electrospinning, and emulsion electrospinning techniques. In Core-shell nanostructures for drug delivery and theranostics. Woodhead Publishing 2018. 325-347.
111 Jiang, S., Chen, Y., Duan, G., Mei, C., Greiner, A., and Agarwal, S. Electrospun nanofiber reinforced composites: A review. Polymer Chemistry 2018, 9(20), 2685-2720.
112 Lu, X., Wang, C., and Wei, Y. One‐dimensional composite nanomaterials: Synthesis by electrospinning and their applications. Small 2009, 5(21), 2349-2370.
113 Xu, Y., Ndayikengurukiye, J., Akono, A. T., and Guo, P. Fabrication of fiber-reinforced polymer ceramic composites by wet electrospinning. Manufacturing Letters 2021.
114 Chen, S., Gao, J., Yan, E., Wang, Y., Li, Y., Lu, H., An, Q., et al. A novel porous composite membrane of PHA/PVA via coupling of electrospinning and spin coating for antibacterial applications. Materials Letters 2021, 130279.
Point 13. Collagen, gelatin, keratin, and silk fibroin are proteins and not polymers. Please, revise it in the manuscript.
Response 13. Thank you for this important comment. Indeed collagen, gelatin, keratin and silk fibroin are proteins and not polymers. We opted to replace the word “polymers” by “materials” in the text.
Point 14. Although many studies consider chitosan a natural polymer, chitosan is a semi-synthetic polymer because it is from the deacetylation of chitin.
Response 14: The reviewer´s comment is right, however we decided to maintain chitosan in the category “natural polymer” in order not to confuse readers.
Point 15. Please, include the correct form of the chitosan units. I recommend a good reference: 10.3390/pharmaceutics13050621.
Response 15. Thanks for the comment. We have corrected the information of chitosan using the recommended reference.
Point 16. What meaning of: HFIP (line 199), DPPH (line 239), FDA (line 319), TFA (line 459), DCM (line 460), SEM (line482), FTIR (line 572), DSC (line 573), Ch (line 575), EDC/NHS (line 669), PBS (line 669- please, include the pH), PLC (line 730).
Response 16. We have written the complete names of the acronyms in the text.
Point 17. The phrase in lines 573-574: "DSC also confirmed that bromelain is encapsulated in nanofibers" is necessary?
Response 17. We agree with your comment. We have deleted this sentence.
Point 18. The English grammar must be revised.
Response 18. The English written was corrected by a native speaker.
Point 19. There is a lack of some Figures/images to comprehend better electrospinning apparatus. Also, the authors could include a Figure/image considering the fibers' applications in treatments of skins burns.
Response 19. We added an image (Figure 1, page 4 line 128) about a schematic representation of an electrospinning apparatus, showing the formation of the Taylor’s cone, for a better comprehension for readers.

Reviewer 4 Report
Dear colleagues,
The article entitled “Electrospun fibers loaded with natural bioactive compounds as a biomedical system for skin burn treatment. A review.” provides a brief description of built electrospun fibers integrating natural bioactive compounds in their structure for an enhanced microbicidal and burned wound healing response. The document interesting, relatively well-written, and new.
Some detailed suggestions can be found below:
Abstract:
I think that you will be expecting the following comment: your abstract is too vague…
“Burn trauma can be caused by various physical or chemical causes.”. - Burn trauma can be caused by …. So many papers and WHO enunciate or even describe in detail the multiple causes of burn injuries.
“The natural healing process consists of varied and complex phases” – instead, enunciate the four stages of natural healing process.
“A wide variety of natural bioactive compounds exists with potential use in dermatology, particularly in the area of wound and burn healing.” – OK. Briefly indicate the main reasons behind this.
“The electrospinning technique is proposed as an inexpensive strategy” – just inexpensive? Not enough. Again, there are multiple advantages of the use of the electrospinning technique towards its applicability in burned wound healing. You are not highlighting the most important ones, particularly in what concerns the regeneration of the injured tissues. Inexpensive compared to that?
“24 publications on this topic were analysed”- why 24? Criteria for paper selection?
You have most of this below, in the introduction. Please, write your abstract in a more attractive manner, leaving the readers eager to know more about it.
Introduction:
Line 43: “polyphenols, micronutrients, enzymes, antioxidants and nutraceuticals” – revise these categories.
Please create a Table 2 showcasing the natural compounds found, similarly as you did in Table 1. Otherwise, a void is created in the content.
Table 2: why 15, and not the 24 that you mentioned earlier? What I am interested as a reader at this point, is to know what is being done for burns, with natural or synthetic compounds, just the electrospinning part. I guess that you have less variations of the technology with natural compounds than with synthetic ones (studied for longer), but it would give an idea of what could be done in the near future. I understand that you may be dissecting the electrospinning process first, but in the way that you wrote you are confusing the readers. I would also like to know about fiber orientation, to have an idea of the final aspect of the mat-derived wound dressing. Is it always random, with basic collector?
Biological Effects of Electrospun-Fibers: you can just write Biological effects, or intended biological effects.
Line 148. Very well. You can add the most frequently found polymers in your research, to justify the following sections where you highlight some of them, and add further explanations for this fact.
Lines 257-258: “Polylactide or polylactic acid is a biodegradable thermoplastic polyester derived from sources such as corn starch, cassava starch and sugar cane.” – the monomer yes. But PLA is synthetically produced… Please correct this throughout the manuscript.
Line 413: please correct the title.
Section 5.3.2. Plant extracts: this general section is misplaced. You should describe first in general categories of natural compounds: polymers, plant extracts, etc., and then describe concrete examples.
Conclusions:
PLC or PCL?
CH/l-arginine (CH-Arg): chitosan is one compound, arginine another…
You highlight PCL. You should highlight as well selected natural compounds as the most promising ones.
“They may therefore offer effective and economically viable treatment alternatives; however, research needs to be broadened and deepened to obtain more conclusive results.” – you need to finish stronger. You should name some of these potentially effective and economically viable treatment alternatives, and then reinforce the advantages of the electrospun mats towards them. Plus, you should give further hints of what exactly is left to improve, so that these potential treatment options can be closer to the intended clinical application.
Author Response
Reviewer 4
Dear colleagues,
The article entitled “Electrospun fibers loaded with natural bioactive compounds as a biomedical system for skin burn treatment. A review.” provides a brief description of built electrospun fibers integrating natural bioactive compounds in their structure for an enhanced microbicidal and burned wound healing response. The document interesting, relatively well-written, and new.
Some detailed suggestions can be found below:
Point 1. Abstract:
I think that you will be expecting the following comment: your abstract is too vague…
“Burn trauma can be caused by various physical or chemical causes.”. - Burn trauma can be caused by …. So many papers and WHO enunciate or even describe in detail the multiple causes of burn injuries.
“The natural healing process consists of varied and complex phases” – instead, enunciate the four stages of natural healing process.
“A wide variety of natural bioactive compounds exists with potential use in dermatology, particularly in the area of wound and burn healing.” – OK. Briefly indicate the main reasons behind this.
“The electrospinning technique is proposed as an inexpensive strategy” – just inexpensive? Not enough. Again, there are multiple advantages of the use of the electrospinning technique towards its applicability in burned wound healing. You are not highlighting the most important ones, particularly in what concerns the regeneration of the injured tissues. Inexpensive compared to that?
“24 publications on this topic were analysed”- why 24? Criteria for paper selection?
You have most of this below, in the introduction. Please, write your abstract in a more attractive manner, leaving the readers eager to know more about it.
Response 1: the reviewer´s comment is right. We have restructured the abstract in a more attractive manner.
Introduction:
Point 2. Line 43: “polyphenols, micronutrients, enzymes, antioxidants and nutraceuticals” – revise these categories.
Response 2: The examples were redefined to “polyphenols, vitamins, minerals, fatty acids, proteins, peptides, probiotics, etc” using as reference Wen et al. [117].
- Wen, P., Zong, M. H., Linhardt, R. J., Feng, K., and Wu, H. Electrospinning: A novel nano-encapsulation approach for bioactive compounds. Trends in Food Science & Technology 2017, 70, 56-68.
Point 3. Please create a Table 2 showcasing the natural compounds found, similarly as you did in Table 1. Otherwise, a void is created in the content.
Response 3: We appreciate your comment. The Table 4 the information that you request.
Point 4. Table 2: why 15, and not the 24 that you mentioned earlier? What I am interested as a reader at this point, is to know what is being done for burns, with natural or synthetic compounds, just the electrospinning part. I guess that you have less variations of the technology with natural compounds than with synthetic ones (studied for longer), but it would give an idea of what could be done in the near future. I understand that you may be dissecting the electrospinning process first, but in the way that you wrote you are confusing the readers. I would also like to know about fiber orientation, to have an idea of the final aspect of the mat-derived wound dressing. Is it always random, with basic collector?
Response 4: The reviewer´s comment is correct. The Table 2 summarizes the 24 publications analyzed, and not 15 as mentioned. The error was corrected (Page 5, line 183). The alignment of the fibers in the collector is random; however, methods such as rotation-al, magnetic, gap or post-drawing are being studied to induce a more ordered alignment in order to expand the mechanical properties and improve a variety of physical properties [118]. This information was added in the text, and also the following reference:
- Robinson, A. J., Pérez-Nava, A., Ali, S. C., González-Campos, J. B., Holloway, J. L., and Cosgriff-Hernandez, E. M. Comparative analysis of fiber alignment methods in electrospinning. Matter 2021, 4(3), 821-844.
Point 5. Biological Effects of Electrospun-Fibers: you can just write biological effects, or intended biological effects.
Response 5: The title of Table 2 was modified to “Table 2. Characteristics of electrospun fibers with intended biological effects from 24 articles reviewed.”
Point 6. Line 148. Very well. You can add the most frequently found polymers in your research, to justify the following sections where you highlight some of them, and add further explanations for this fact.
Response 6. We have added the information requested in the text.
Point 7. Lines 257-258: “Polylactide or polylactic acid is a biodegradable thermoplastic polyester derived from sources such as corn starch, cassava starch and sugar cane.” – the monomer yes. But PLA is synthetically produced… Please correct this throughout the manuscript.
Response 7. We appreciate your comment. The polymer “Polylactide” was moved to the synthetic materials section. In addition, it was modified in Table 3.
Point 8. Line 413: please correct the title.
Response 8. The heading “5. Polymers used for the production of electrospun fibers” was effectively wrong, furthermore it was replaced by “5. Natural bioactive compounds used in electrospun fibers.”.
Point 9. Section 5.3.2. Plant extracts: this general section is misplaced. You should describe first in general categories of natural compounds: polymers, plant extracts, etc., and then describe concrete examples.
Response 9. The natural compounds were categorized by their bioactivity, and not by their type (polymers, plant extracts, oils, etc.). However, we agree that the name “Plant extract” is not appropriate, so it was replaced by “Memecylon edule extracts”.
Conclusions:
Point 10. PLC or PCL?
Response 10. You are right, the correct acronym is PCL. We have corrected this error through the text.
Point 11. CH/l-arginine (CH-Arg): chitosan is one compound, arginine another…
Response 11. We deleted this paragraph in the conclusion.
Point 12. You highlight PCL. You should highlight as well selected natural compounds as the most promising ones.
Response 12. You are right. We added this sentence in the conclusion: Given their similarity to physiological extracellular collagen matrix, natural polymers are demonstrated to be attractive materials for biomedical applications as they have the advantage of biological recognition and potential bioactive behavior proliferation. In fact, they are intrinsically biocompatible and biodegradable, and degradation products are well tolerated and metabolized by the human body. Among the synthetic materials, PCL was the most widely used (6 publications), probably because of its favorable physicomechanical properties for electro-spinning, its biodegradability and its cellular biocompatibility
Point 13. “They may therefore offer effective and economically viable treatment alternatives; however, research needs to be broadened and deepened to obtain more conclusive results.” – you need to finish stronger. You should name some of these potentially effective and economically viable treatment alternatives, and then reinforce the advantages of the electrospun mats towards them. Plus, you should give further hints of what exactly is left to improve, so that these potential treatment options can be closer to the intended clinical application.
Response 13. We have completely changed the conclusions as your recommendations.

Reviewer 5 Report
- Why did you choose period from 2006 to 2021 for review?
- Please characterize pros and cons of electrospinning. Explain why electrospinning gives advantages over existing methods of wound treatment? What are the limitations of using electrospinning method?
- Why did you choose encapsulated natural biological active compounds instead of synthetic non-biological compounds?
- Why was the choice to include only burns in the analysis and not diabetes or other difficult-to-heal wounds?
- Also, you can mention combining electrospun materials with other active compounds e.g. sorbents.
- Figure 1. Mark in figure where is a, b and c accordingly?
- Define “O/W” (line 120).
Author Response
Reviewer 5
Point 1. Why did you choose period from 2006 to 2021 for review?
Response 1. This period of time selected is a consequence of the WOS applying the keywords "nanofibers"; "electrospun"; "skin" and "burns" and selecting the publications that address the topic of this review. The results obtained after this search covered the years 2006 to 2021.
Point 2. Please characterize pros and cons of electrospinning. Explain why electrospinning gives advantages over existing methods of wound treatment? What are the limitations of using electrospinning method?
Response 2. Thanks for your comments. Regarding the pros and cons of the use of electrospinning for the treatment of skin burns, we proceed to deliver that information in the introduction section. It was necessary to add a new reference for this:
107 Khorshidi, S., Solouk, A., Mirzadeh, H., Mazinani, S., Lagaron, J. M., Sharifi, S., and Ramakrishna, S. A review of key challenges of electrospun scaffolds for tissue‐engineering applications. Journal of tissue engineering and regenerative medicine 2016, 10(9), 715-738.
Point 3. Why did you choose encapsulated natural biological active compounds instead of synthetic non-biological compounds?
Response 3: At present there is great interest in natural compounds with potential applicability in the medicine area. Natural products have a variety of components, including alkaloids, flavonoids and ter-penoids that are abundantly present in different sources of medicinal plants, marine life, fruits and vegetables [108]. In the case of treatments for burns, the importance lies in the capacities they could possess both in the control of infections by common and resistant microorganisms and in the regeneration of the affected tissues, having in both cases low adverse effects.
This information was added on page 1 lines 40-45. In addition, the following reference was added:
108 Mohammadinejad, R., Madamsetty, V. S., Kumar, A., Varzandeh, M., Dehshahri, A., Zarrabi, A., Ramakrishna, S., et al. Electrospun nanocarriers for delivering natural products for cancer therapy. Trends in Food Science & Technology 2021.
Point 4. Why was the choice to include only burns in the analysis and not diabetes or other difficult-to-heal wounds?
Response 4. Diabetes wounds or other hard-to-heal wounds were not include for the purpose of writing a focused review, and not too broad topic.
Point 5. Also, you can mention combining electrospun materials with other active compounds e.g. sorbents.
Response 5. We have included only some active compounds with the purpose not extend too much the review.
Point 6. Figure 1. Mark in figure where is a, b and c accordingly?
Response 6: Figure 1 was modified and the letters a, b and c were added accordingly.
Figure 1. Types of electrospun fiber: a) simple; b) blend; c) core-shell.
Point 7. Define “O/W” (line 120).
Response 7. O/W is the acronym Oil/Water. We added this information in the text.

Round 2
Reviewer 2 Report
The review has been improved.
Reviewer 4 Report
Well done, congratulations.